# Obérisk: Cybersecurity Requirements Elicitation through Agile Remote or Face-to-Face Risk Management Brainstorming Sessions

**Stéphane Paul [1,*]** , **Douraid Naouar [2,*]** and **Emmanuel Gureghian [1]**

[1]  Thales Research & Technology, 91767 Palaiseau, France; emmanuel.gureghian@thalesgroup.com
[2]  Chair of Naval Cyber Defense, Ecole Navale, 29160 Lanvéoc, France
*  Correspondence: stephane.paul@thalesgroup.com (S.P.); douraid.naouar@ecole-navale.fr (D.N.)

**Abstract:** Cyberattacks make the news daily. Systems must be appropriately secured. Cybersecurity risk analyses are more than ever necessary, but . . . traveling and gathering in a room to discuss the topic has become difficult due to the COVID, whilst having a cybersecurity expert working isolated with an electronic support tool is clearly not the solution. In this article, we describe and illustrate Obérisk, an agile, cross-disciplinary and Obeya-like approach to risk management that equally supports face-to-face or remote risk management brainstorming sessions. The approach has matured for the last three years by using it for training and a wide range of real industrial projects. The overall approach is detailed and illustrated on a naval use case, with extensive feedback from the end-users. We show that Obérisk is really time-efficient and effective at managing risks at the early stages of a project, whilst remaining extremely low-cost. As the project grows or when the system is deployed, it may eventually be necessary to shift to a more comprehensive commercial electronic support tool.

**Keywords:** risk management; EBIOS; face-to-face; remote; agile; brainstorming; inter/cross-disciplinary; obeya; posters; sticky notes; cybersecurity requirements; naval/ship use case





## 1. Introduction

A major trend in system and software development relates to DevOps, closely followed in time by DevSecOps. DevOps is about breaking the siloes between development, quality assurance and operations, to ensure continuous integration and delivery. A short history of DevSecOps [1] recalls that the first spark for DevOps was given in 2009, by a Flickr talk [2], promoting the lowering of risks of change with automation tools, and an adequate enterprise culture. In 2015, DevSecOps put the spotlights on security, recalling that security should be given precedence over new features [3], and stressing that security is not under the responsibility of the sole security experts. These approaches have been immensely popular, and commercially successful, especially related to automation. Indeed, some 100% technical profiles are now tagged DevOps as soon as they master the latest web automation tools. Likewise, most DevSecOps commercial offers relate to process automation (e.g., [4–6]), whilst little is commercially available towards facilitating collaborative risk management. Where are the soft skills and enterprise culture parts mentioned in the Flickr talk? Where is the "everyone is responsible for security" of Shannon Lietz [7], in particular when it comes to the involvement of the executive management and the board of directors? Where are the methods and tools that can leverage that collaborative spirit in decision making, especially concerning the elicitation of system cybersecurity requirements?

This article proposes Obérisk, an Obeya-like tooled-up approach to system risk management that leverages collaboration between all stakeholders. These include system developers, but also end-users, sponsors or public authority stakeholders. It can be run in face-to-face meetings or in remote brainstorming sessions to allow for the early elicitation

of cybersecurity requirements, typically as a one shot process (e.g., for a bid) or as an iterative process (e.g., between development sprints). Naturally agile, the approach allows iterations to cope with any change or deep-dive into a specific issue. It delivers fast, i.e., within 10 to 14 h for a complete study. The final report is a model of conciseness (approx. 14–17 pages, all included), and it is graphical so that it can be shared and easily read by all. It includes all the cybersecurity requirements, together with all the justifications of why they are needed and where they should be applied. In short, it offers a comprehensive shared view of the system-under-study and its cybersecurity landscape for everyone involved.

A first paper was already published about this Obeya-like approach [8], but at the time it only supported face-to-face meetings. Incidentally, it was not yet called Obérisk, and the French EBIOS Risk Manager [9] method, on which it is based, was not yet available in English. The COVID pandemic was a game-changer. Cybersecurity risk assessments are more than ever necessary [10] but traveling and gathering in a room to discuss the topic has become difficult, if not impossible. This article presents the overall approach but focuses in particular on the new features supporting remote brainstorming sessions. A complete risk assessment and treatment cycle can now be run in five to seven teleconferences, with more participants in the loop, ensuring a better coverage of the different stakes. With remote sessions, time-efficiency is increased, because back office work is reduced compared to face-to-face meetings. All the while, the work remains fun, and a major vector for team building.

The paper comprises five sections and two appendixes. Section 2 justifies why a new easy, agile and more collaborative approach is needed for risk management. It also highlights its novelty and benefits with respect to existing methods. Section 3 describes the methodological approach of this research, whilst Sections 4 and 5 present and discuss the resulting Obérisk framework. Appendix A provides an overview of the EBIOS Risk Manager method. It is recommended that readers not familiar with this risk management method read the appendix before reading Sections 4 and 5. Appendix B provides a listing of all the risk management datasheet templates proposed in the Obérisk framework.

## 2. Literature Review

Risk management has long been recognized as a best practice in industry [11,12]. It has been designated as one of the eight main areas of the Project Management Body of Knowledge (PMBOK) [13] by the Project Management Institute, and one of the key system features desired by stakeholders in the INCOSE Vision 2025 [14]. Even though risk management is of crucial importance for industrial project success, very few models explicitly relate risk management with agile development processes. One reason is that extending the agile model with additional development practices is a highly controversial problem. It is often argued that one of the most challenging impediments to developing the agile model relates to the following question: "How can one merge agile, lightweight processes with standard industrial processes without killing agility?" [15,16]. This is also what emerges from recent publications dealing with projects using agility [17,18]. Many large organizations show a genuine interest in using an agile model, but they are hesitant due to the experienced scalability issues. Essentially, they are challenged by the lack of guidelines for building up the agile process according to their needs, where one of the missing building block is risk management.

There are many cybersecurity risk management processes and methods [19,20] around the world, each with its own strengths and shortcomings [21,22]. They are proposed by standardization bodies (e.g., ISO [23], IEC [24]), by State Ministries/National Security Agencies (e.g., American NIST [25], German BSI [26], French ANSSI [9], Spanish Ministry of Finance and Public Administration [27]), by universities (e.g., Carnegie Mellon [28]) and even industry (e.g., Mozilla [29]) or associations (e.g., CLUSIF [30]). Amongst these, only a few processes (e.g., ISO 3100 [31]) and methods (e.g., the EBIOS Risk Manager [9], OCTAVE ForTE [28] or RRA [29]) really encourage joint work as a key factor of success. None provide practical details on how to facilitate brainstorming sessions involving general engineers,

management or legal staff, in addition to security experts. The need to equip these methods is only greater, and several proposals have been suggested, e.g., [32–34]. These tools are software solutions based on knowledge bases, allowing easy access to threats or security measures present in the standards and literature [35]. They also allow automating specific tasks. However, these tools do not help end-users understand the fundamentals of a method, and even less address the required soft skills. Many people imagine that buying a tool will help them correctly implement the method. It is a lure. Software solutions support many options, offer a wide range of databases, and are usually highly configurable. Adapting them to your needs requires some significant experience and does not always allow for a proper understanding and consideration of each other's concerns.

The topic of soft skills, and the underlying collaboration between actors, recently emerged as a need in project management at large, and for risk management in particular [36–39]. According to Crawford et al. [40], employers are beginning to focus more on behaviours or what are often referred to as soft skills in addition to, or in some cases rather than, hard technical skills. Marly Monteiro de Carvalho et al. [41] states that the soft side of risk management explains 10.7% of the effect on project success. Likewise, with respect to scholarship, Söderlund et al. [42] argue for the need of combining the hard and soft issues of management. The absence of soft skills has produced a lack of understanding of the social dimensions of management and organization. However, the balance is critical, as too much focus on the soft side of management hinders the good use of the techniques and analytical tools available to make wise decisions.

Ignorance, cost and complexity are major factors that prevent the systematic rollout of risk management practices [43]. For this reason, we assert that a cheap, easy, understandable and quick approach to risk assessment is needed. It must allow teams and stakeholders to conduct risk assessments within an agile process, experiment and become aware of the cybersecurity concerns of the field very early in the development lifecycle. Later, to deepen the identified issues, the analysts may use more systematic and comprehensive approaches. We propose an approach that complies with the EBIOS—Risk Manager method (as a pledge of seriousness and credibility)—, emphasizing the maieutic and the ergonomics, especially in remote conditions, the speed of setting up and implementation, and of course low costs, in terms of time and direct expenses. The approach leverages and magnifies the soft skills of the risk assessment facilitator.

## 3. Method

The original idea for Obérisk arose from a draft release of ANSSI's Agility and Digital Security guide [44]. Draft 0.42 of the guide, dated July 2017, proposed a giant paperboard canvas to host risk management sticky notes, organized in seven categories: (1) business assets and their security needs, in terms of confidentiality, integrity, and availability; (2) risk sources and their characterization in terms of profile, motivation, objectives, and skills; (3) feared events and their impacts, with their severity; (4) system components, including organizations and human resources and their vulnerabilities; (5) existing security measures; (6) threat scenarios with their likelihood of occurrence; (7) the risks, with their criticality and proposed countermeasures. This canvas did not scale for real studies. It was removed from the final version of ANSSI's guide. However, this canvas was appealing in terms of information structuring. Thales derived three Microsoft PowerPoint™ A0-format posters from this canvas to support 1-day training sessions to risk management techniques for internal system engineers and architects. The Thales internal training has been regularly provided 15 to 20 times per year since mid-2016, except during the COVID period, so we had a chance to collect considerable feedback on the use of the posters and improve them over time. The choice of PowerPoint™ and printed posters was made to provide a low-cost and easy deployable solution compared to on-the-shelf electronic tools managing sticky notes, e.g., iObeya [45] or MURAL [46].

Table 1 summarises the different contexts during which the Obérisk approach was used, and feedback collected to improve the approach.

**Table 1.** Obérisk validation contexts and participants (training and use cases).

| Exercise Type | Dates and Occurrences | Number of Participants | Profile of Participants |
|---|---|---|---|
| Thales internal cybersecurity training: engineering, risk management and architecting | Mid-2016 to end 2020 94 sessions | 16–20 per session, for a total of ≈ 1700 | System engineers; system architects; integration, verification and validation engineers |
| Risk management course at engineering school | 4 sessions, 2018–2021 | 24–28 per session, for a total of ≈ 100 | Master 2 students |
| Thales internal risk management training | 3 sessions, 2019–2020 | 7–30 per session, for a total of ≈ 50 | Cybersecurity specialists |
| Thales Low Altitude Air Traffic Management use case | Mid-2018 | 3 | Senior architects; domain experts |
| Thales Air Traffic Geofencing use case | End 2018 | 3 | Senior architects; domain experts |
| Thales IoT for SCADA use case | Early 2019 | 7 | System engineers and architects; domain experts; cybersecurity expert |
| Thales Railway Signalling use case | Mid-2019 | 12 | System engineers and architects; domain experts; developers; managers; cybersecurity expert |
| Thales Cloud-Computing use case | Mid-2019 | 2 | System engineers and architects; cybersecurity expert |
| Thales In-Flight Entertainment use case | End 2020 | 5 | System engineers and architects; cybersecurity experts |
| FORESIGHT H2020 naval use case | End 2020 to mid-2021 | 7 | PhD students; domain and cybersecurity expert |
| FORESIGHT H2020 airport landside use case | Early 2021 to mid-2021 | 3 | Manager; domain expert; cybersecurity experts |
| FORESIGHT H2020 power grid use case | Early 2021 to mid-2021 | 9 | System engineers and architects; domain experts; cybersecurity experts |
| College of Practitioners of the Club EBIOS | End 2020 to early 2021 | 5 | Senior risk management practitioners |

The posters were further improved and expanded for a 16 h risk management course for Cybersecurity Master 2 students of a French engineering school [47]. To allow for appropriate scaling, we provided six A0-format posters. These posters have now been used successfully for four consecutive years in that school. Starting from 2019, the Obérisk posters were also used on a Thales internal risk management training for cybersecurity specialists. The panel of training participants, thus, extends from students to professionals, and from generalists to cybersecurity specialists.

Training professionals and students was an excellent opportunity to improve the posters. Over the years, we had the opportunity to meet some two thousand trainees and observe their reactions and ways of reifying risk assessment information on the posters. Training also provided an opportunity to observe collaborative work, as the trainees are requested to work on a case study in small groups of 3 to 5 trainees. Beyond observation (i.e., expert opinion on participant behaviour and actual work progress), feedback was also explicitly requested from the trainees at the end of each workshop and at the end of the training sessions.

One of the major topics addressed when analysing the training return on experience was related to terminology. Methods usually introduce fancy vocabulary about risks. For example, the EBIOS Risk Manager method [9] introduces the following concepts, all of

them directly related to the concept of risk: strategic scenarios, attack routes, operational scenarios, attack paths and risk scenarios. These terms are often confusing for neophytes, and subject to long debates amongst experts. Significant work was performed on the Obérisk terminology to simplify the risk management terms without introducing nonsense. For this work, the participation of cybersecurity specialists was critical since they did not report any misinterpretations due to the terminology simplifications.

However, training on a predefined case study provides little variety in terms of business constraints and mixes educational concerns with efficiency concerns. So, in parallel to training, running risk assessments with the posters on many small Thales industrial projects (cf. middle part of Table 1) provided us with a much richer set of operational constraints. For example, some projects had a very rich set of already deployed security measures that needed to be accounted for. Some projects had detailed architecture schemas readily available, while other projects related to futuristic ideas, with a completely undefined or speculative architecture. Some projects could be handled with two persons during the brainstorming sessions, while other projects required up to twelve persons to provide a good domain knowledge coverage. However, all these projects had a common point: participants of these use cases had never run a risk assessment before. In less than two days (i.e., 5 to 7 two-hour workshops), the objective was to effectively run the risk assessment and bring the participants to a sufficient level of cybersecurity maturity to allow them to be independent. Each project proved to be an opportunity to fine-tune the posters to the project's operational constraints. Data were essentially collected through observation. Nonetheless, after some sessions, explicit surveys were also run, with the following 10 questions:

1. Overall, how would you rate the risk assessment brainstorming sessions?
2. Did the set of workshops meet its objectives?
3. What did you like about this event?
4. What do you think should be improved?
5. Do you feel you can now develop/maintain the analysis by yourself (as a team)?
6. Do you think the approach would be applicable to other division system/product/equipment?
7. Do you feel you could now run a new risk assessment by yourself on another system/product/equipment?
8. The approach was developed to be specifically applicable at bid time, for an initial security analysis. Would you agree that this would be an adequate setting for this approach?
9. How likely is it that you would recommend this approach to Thales colleagues?
10. Any other comment?

Some of the survey answers obtained on the Thales Railway Signalling use case (cf. middle part of Table 1) are shown in Figure 1. This survey was one of the most significant surveys, as there was a stable set of 12 participants throughout the whole set of workshops.

More recently, the COVID pandemic required that the risk management sessions be organized remotely. This did not have a strong impact on the layout of the datasheets, but it forced us to organize the elements in multiple layers. Indeed, when the datasheets were printed as A0-formatted posters before being physically hanged in a room, the actual design of the datasheets is unimportant. However, when the datasheets are shared and filled in online during teleconferences, the undesired selection of background elements can rapidly become very annoying. For the three risk management use cases on the European H2020 FORESIGHT project [48] (cf. bottom part of Table 1), this constraint led to the extensive development of PowerPoint™ templates. Each workshop now has its dedicated set of templates (cf. Appendix B for the complete list). Each template provides a canvas for the risk management data, defining what type of data must be collected, and characterizing them based on their location on the canvas.

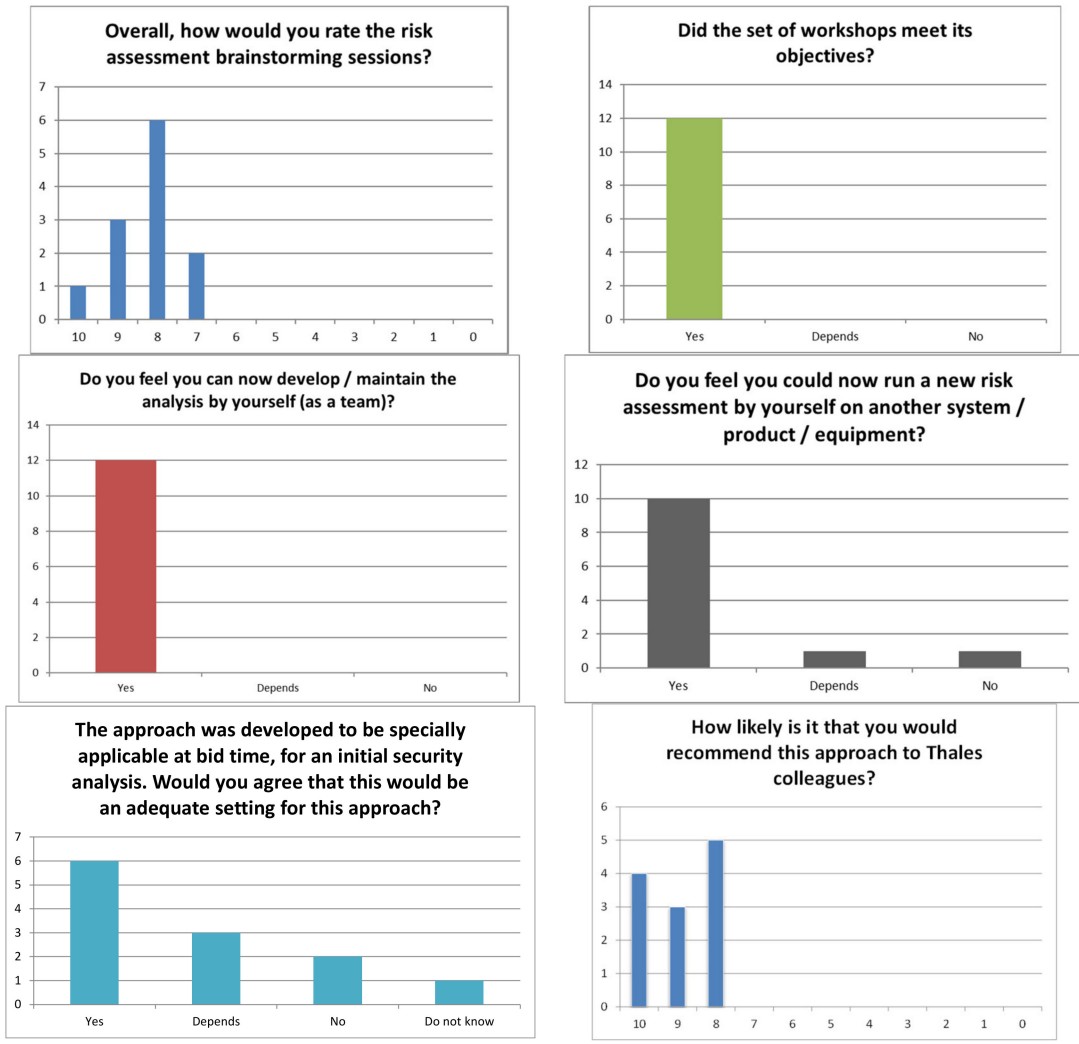

**Figure 1.** Extract of railway signalling use case survey results.

Finally, the Obérisk roadmap to maturity climaxed when the project was submitted to the College of Practitioners of the Club EBIOS [49]. The datasheets were reviewed and further improved by five senior risk management practitioners external to Thales. A user guide was also developed (in French only at the time of writing this article).

## 4. Results

Obérisk is a tool-up approach to the French EBIOS Risk Manager [9] method. Understanding the pros and cons of Obérisk requires some knowledge of the method. This paper does not aim to present or promote the EBIOS Risk Manager method; however, we provide an overview of the method's key features in Appendix A to allow for a self-contained article. In this section, we also recall the main goals of each workshop before presenting the Obérisk features related to that workshop.

Obérisk is made available in the form of PowerPoint™ slides under the CC BY-NC-SA (i.e., Creative Commons Attribution + Non-Commercial + Share Alike) licence. The Obérisk package includes a set of datasheets, a user guide and a full-blown example. The datasheets and example are in English. The user guide is in French. They can be found on the Club EBIOS forum [50] or on Research Gate [51]. The datasheets are presented in detail below.

The set of Obérisk PowerPoint™ datasheets comprises 18 different layouts, covering all the EBIOS-Risk Manager method workshops. This set of slides includes 11 standard sheets, sufficient to perform a complete risk management study, and 7 alternative sheets

(see Appendix B for the full list). The alternative sheets are essentially designed to cope with scalability requirements, or ergonomic preferences.

All the Obérisk datasheets are designed using PowerPoint™ templates. This is particularly useful when filling the datasheets during remote brainstorming sessions using teleconference support tools. Indeed, only the sticky notes in the foreground can be selected and modified with the mouse during the session. The basic layout of the datasheet cannot be tampered with. Additionally, templates allow adding a new sheet with a given layout, or changing the layout of a given page, if a scalability issue arises during the workshop. In short, Obérisk leverages the familiarity of the end-users with the Microsoft Office automation tools.

In this article, the content of the datasheets and the way to use them are described for each workshop. After a generic presentation, the use of the datasheets is illustrated through their application in the FORESIGHT project [48] to perform the risk analysis of a commercial passenger ship. The FORESIGHT project proposes a federated cyber-range solution to improve the preparation of cybersecurity professionals by establishing an ecosystem of realistic networked training and simulation platforms. The Chair of Naval Cyber-Defence [52] is the designer and responsible for the naval cyber-range. Thales' involvement in this project and its collaboration with Chair of Naval Cyber-Defence, allowed us to practice Obérisk on an unclassified, but complex and demanding project, with an access to a wide range of profiles: doctoral students in cybersecurity, automation experts, experts in the maritime field, platform engineers... To retrieve the complete analysis of the maritime use-case discussed herein, we invite everyone to visit the Club EBIOS forum [53].

### 4.1. Workshop n°1: Framing and Security Baseline

The Framing and Security Baseline is the first workshop of the EBIOS Risk Manager method. Its goals are to frame the system-under-study, identify its missions and security needs, and start building a cybersecurity engineering strategy. Its participants should be a top manager, a domain expert, the Chief Information Security Officer (CISO), and the Chief Information Officer (CIO). The expected outputs are some framing elements, e.g., study objectives, roles and responsibilities, the domain and technical perimeter, including business/operational assets, the feared events (also known as hazards) and their severity, and the Minimal Set of Security Controls (MSSC) to be applied. Workshop n°1 is supported by three standard Obérisk datasheet layouts and four alternative layouts.

#### 4.1.1. Definition of the Study Objectives, System Missions, and Asset Owners

The first Obérisk datasheet for workshop n°1 is called the *Framing and security baseline*. It is numbered per default one of three (1/3) in the top right corner since three sheets are normally required to cover the first EBIOS Risk Manager workshop. If alternative sheets are used, the numbering will need to be updated accordingly. The first Obérisk datasheet is split into three sections (cf. Figure 2). The first section allows capturing:

- The study objectives within pinkish sticky notes. By default, the datasheet allows room for three different objectives, but it is possible to have less.
- The missions of the system-under-study within blue sticky notes. Per default, the datasheet allows room for only two different objectives. This can be reduced to a single item if applicable. It is essential to keep the number of missions low; otherwise, some confusion might arise with the elicitation of the business assets (see below).
- Optionally, the analysis timeframe. The EBIOS Risk Manager method comprises two cycles: the strategic cycle and the operational cycle. The expected duration of those cycles can be documented here. Although they are seldom filled in, these fields are provided here to ensure compliance with the method. In practice, these durations are difficult to assess, especially when the strategic and operational cycles relate to different stakeholders, e.g., the customer versus the system's end-user.

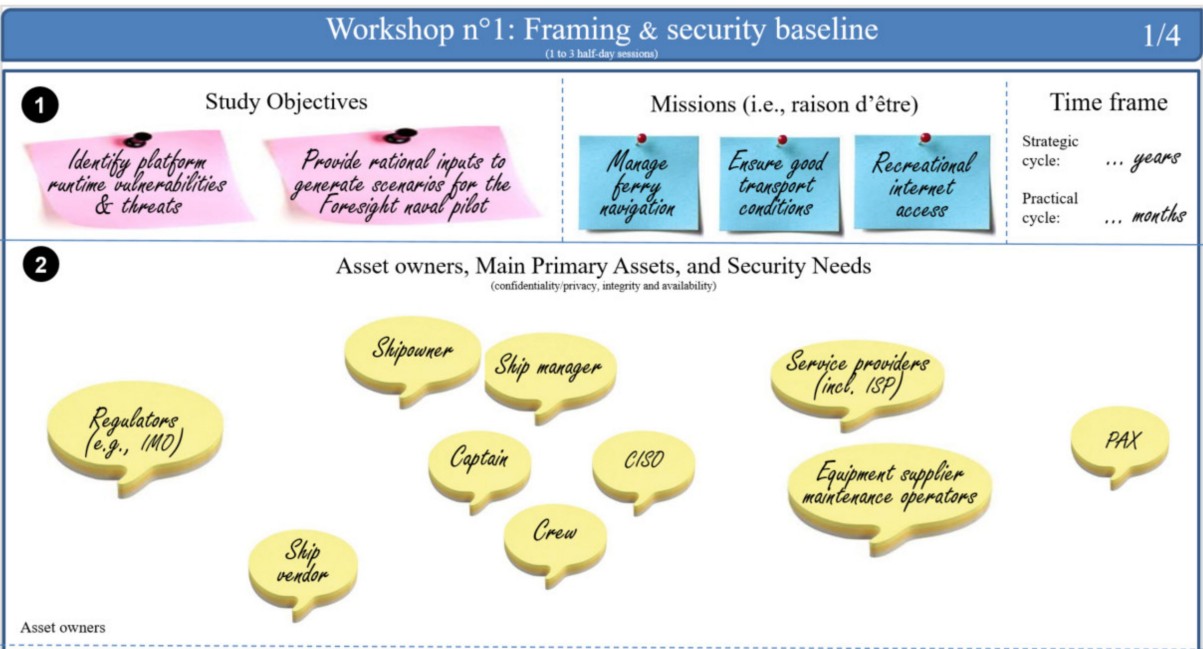

**Figure 2.** Naval use case—study objectives, system missions and key asset owners.

The second section is dedicated to assets and asset owners. In the upper part of this section, the asset owners are captured in yellow cartoon bubbles to symbolize those who have their say on the matter. The datasheet allows sufficient space to group the asset owners per type. For example, the different partners, sub-contractors, suppliers, etc., can be grouped in small packs.

### 4.1.2. Illustration in a Naval Setting

As shown in Figure 2, our study applied to the FORESIGHT platform has two objectives: (i) to identify vulnerabilities and threats related to the platform's operational functioning; (ii) to provide analytical data to generate scenarios for the FORESIGHT naval pilot. It is noteworthy that, due to these objectives, workshop n°5 (cf. Section 4.5), dedicated to risk treatment, will not be performed: indeed, the goal is to provide risk treatment through ship crew training on the FORESIGHT cyber range.

With respect to the missions, among all those that the system must carry out, we decided to select three that we could refer to as the primary missions: (i) *manage ferry navigation*, (ii) *ensuring good transport conditions* and (iii) *recreational internet access*.

For asset owners, we identified many profiles starting from *service providers, equipment suppliers, maintenance operators* and the *regulators* (to which we should refer when it comes to certification), up to the persons having access to the ship: *crew, ship manager, ship owner* and passengers (*PAX*).

Lessons learnt: This first work was essential because it allowed us to keep our objectives in mind for the rest of the workshops and see if the point of view with which we approached the workshops was still in line with the desired work. Furthermore, the part on the asset owners allowed us to identify the people who will ultimately have to apply or enforce the security measures and even have a word to say about the security measures identified at the end of the study. We need to keep in mind that each of these asset owners had their concerns, and that taking them into account, or knowing their existence, was key in the discussions.

### 4.1.3. Definition of Primary Assets

In the lower part of the second section (cf. Figure 3), the business/operational assets are captured within light blue sticky notes and positioned on a security need axis, repre-

sented by a horizontal line pointing to the right. The Obérisk datasheet allows for space above and below the security need axis. It is recommended to locate process-type assets, a.k.a. services or functions, above the axis and locate informational assets below the axis. Process-type assets will usually be concerned by integrity (I) and availability (A) needs, while informational assets may also be concerned by confidentiality (C) and privacy (P) needs. The rising security needs are represented by the asset position alongside the axis and its simple colour code, from green to orange and then red. Traditionally, risk management methods use scales to capture security needs, with precise steps within those scales. Here, there are no specific values. What is important is the relative position of an asset compared to the other assets and its absolute position on the axis. A business asset positioned on the right of another one has a stronger security need and vice-versa. The security need axis is used to represent the needs related to all security criteria, i.e., confidentiality (C), integrity (I), and availability (A). The security criterion at play is captured by writing the first letter of the security criterion in the upper right corner of the asset stick-note, as shown in Figure 3. If the security need levels of a given business asset are identical across multiple criteria, those criteria can be written on the same sticky note. On the contrary, if the security need levels of a given business asset are different across multiple criteria, multiple sticky notes need to be created and located at different places along the axis.

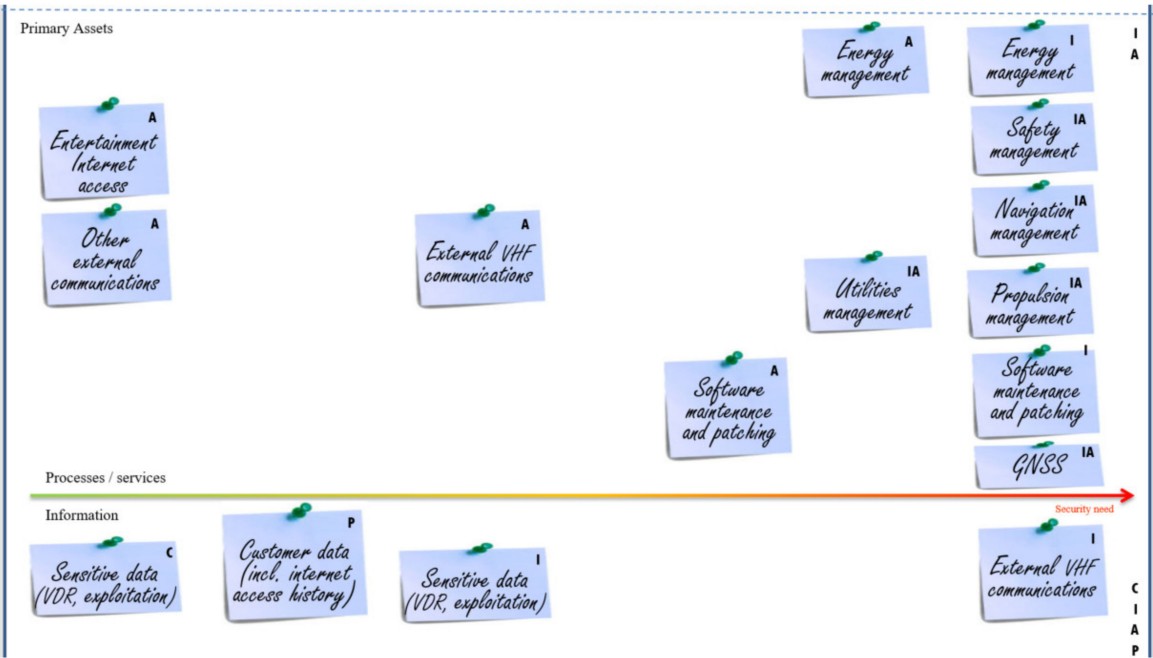

**Figure 3.** Naval use case—key business assets and their security needs.

To avoid the redundant capture of security needs on processes and their related data and limit the number of sticky notes, it is recommended to work essentially on process-type assets. Indeed, if a process has integrity and availability needs, and this process uses or produces some information, then the same integrity and availability needs will be subsumed on the information. The contrary is untrue. Thus, informational assets will essentially be used to capture confidentiality and privacy needs. Exceptions to this recommendation include assets such as logs, which need to remain of integrity and available independently of the process which created those logs.

Optionally, privacy (P) needs may also be managed, but this is a little bit trickier. Indeed, privacy is not usually expressed with a scale. Data is private or is not private: there is no intermediate status. Thus, privacy is not a security criterion such as the CIA criteria. Our recommendation is to locate the business asset sticky notes along the security need axis based on the volume of processed data or data subjects. This stems from the

fact that the General Data Protection Regulation (GDPR) [54] has specific clauses when processing involves a large amount of personal data or affects many data subjects. When a large amount of personal data or a large number of data subjects are involved, the security requirements are stronger by regulation.

### 4.1.4. Illustration in a Naval Setting

In the context of our study, we have identified several business assets. Some are considered processes such as *energy management, safety management, navigation management, propulsion management, software maintenance and patching, entertainment internet access*. Others are considered as informational assets: *external VHF communications, customer data* and other sensitive data, such as the Voyage Data Records *(VDR)*.

In terms of security need assessment, we have determined that the *energy management* asset needs to be of-integrity and highly available, but the availability need is slightly less stringent than the integrity need. Two sticky notes were created to represent this: one on the far right with the integrity need and the other positioned to its left. By contrast, we have determined that *safety management* has its integrity and availability security needs considered equally important: we gathered these two needs on the same sticky note.

Lessons learnt: This work allowed us to clearly identify the primary assets performing the missions of the system defined above and the level of security they required. This allowed internal and external stakeholders to see and understand which features we were dealing with and which are the most critical for the system's proper functioning.

### 4.1.5. Definition of the Existing Regulatory Security Controls

The third and last section of the Obérisk *Framing and security baseline* datasheet is split in two (cf. Figure 4).

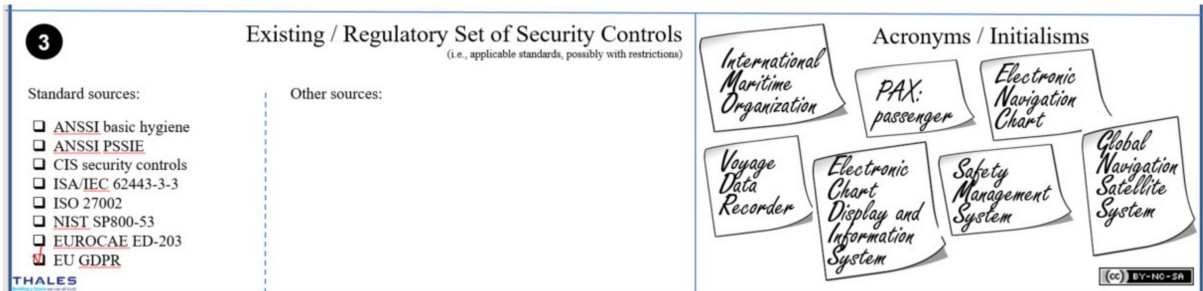

**Figure 4.** Naval use case—security regulations, policies, security standards and abbreviations.

The left part is dedicated to applicable regulations, policies and security standards. The left-most part lists a set of well-known security regulations, policies and security standards. This list needs to be adapted in the slide template, before the brainstorming sessions, with items often used in the domain-under-study. Thus, during the session, it will be sufficient to tick the appropriate items. If the required regulations, policies or security standards are not already on the list, additional ones can be added on yellow sticky notes.

The right part of the third section provides a little space to recall typical acronyms and initialisms used in the domain-under-study. The latter may seem anecdotal, but understanding and sharing the vocabulary of the stakeholders is always a good step towards collaboration and establishing trust with the stakeholders, especially the unaware cybersecurity stakeholders.

### 4.1.6. Illustration in a Naval Setting

As part of our study, the platform respected and applied the European General Data Protection Regulation (EU GDPR). As for the acronyms, most of them come from the naval domain to improve the relevance and comprehensiveness of the concepts defined with the business operators.

Lessons learnt: This first datasheet is intended to be clear and understandable for everyone, whether or not they are involved in the study. Through these first three sections, this datasheet allows us to see the study's objectives and the impacted people, a classification of the primary assets and standards and regulation to which we refer.

### 4.1.7. Hindsight on the First Datasheet

The first Obérisk datasheet is usually sufficient to cope with most risk management studies. If there is not enough space on the datasheet for the sticky notes, the reflex should be to question oneself about the level of abstraction used for the analysis. Typically, if there are more than ten business assets, some business assets may be grouped as a single asset. This is key for rallying the stakeholders. However, some systems may be particularly complex, typically when studying a system of systems, or when the number of stakeholders is very high or when the deployment is scheduled across multiple countries with different mandated security regulations, policies and standards. For these particularly complex cases, Obérisk offers to replace the previously presented datasheet with two datasheets with the same sections, but allowing much more space for stakeholders and applicable regulation, policies and security standards. The space for the business asset is only slightly increased because the risk analysis is normally performed on behalf of a single stakeholder, i.e., the risk analysis sponsor. Therefore, the number of business assets should not increase proportionally to the number of stakeholders.

### 4.1.8. Definition and Assessment of the Feared Events

The second standard Obérisk datasheet for workshop n°1 is called *Business impact assessment*. It is split into two identical sections numbered 2a and 2b (cf. Figure 5). The numbering refers to Section 2 of the first Obérisk datasheet. As explained above, Section 2 of the first Obérisk datasheet is dedicated to assets and their security needs. This datasheet extends this analysis by capturing the consequences of an infringement of those security needs to assess the severity of this breach. Feared events, on pink sticky notes, reify the infringement of the identified security needs. For each security need expressed on the first Obérisk datasheet, a feared event must be created here, with the same label, and its security need must be prefixed with the unary negation symbol (i.e., "¬") [55]. This symbol is a handy shortcut to capture the negation in a very condensed format.

Below each feared event, there is ample space to place the consequences of the security need infringement, captured as impact statements on orange sticky notes. The impacts are located vertically along with a Business Impact Level (BIL) axis, ranging from low impact in green to high impact in red. Traditionally, risk management methods use scales to capture BILs, with precise steps within those scales, e.g., [56]. Here, there are no specific impact level values. What is important is the relative position of an impact compared to the other impacts and its absolute position on the axis. An impact statement positioned above another one has a stronger impact level and vice versa.

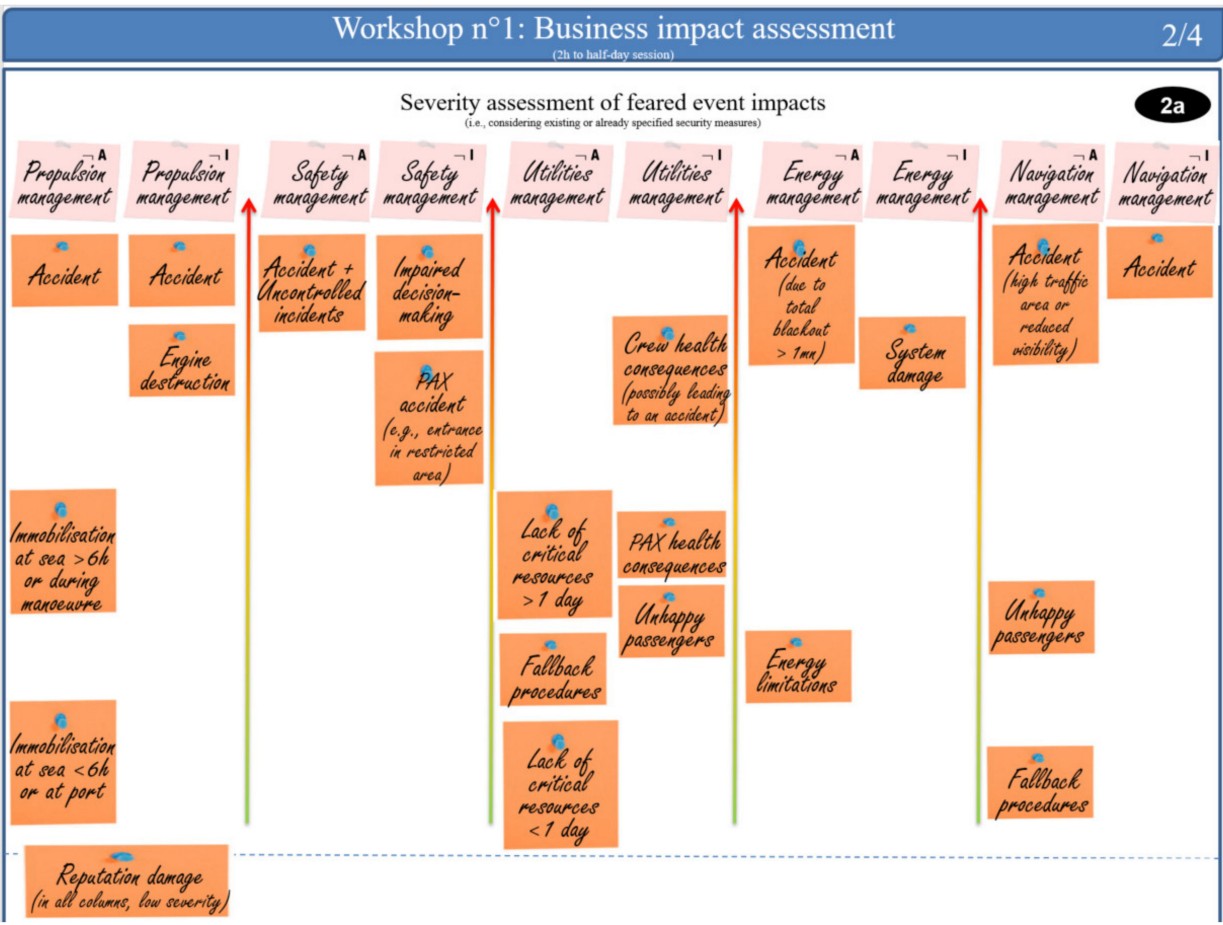

**Figure 5.** Naval use case—business impact assessment.

### 4.1.9. Illustration in a Naval Setting

As shown in Figure 3, we have previously identified and ranked key assets in terms of security needs. Here, we analysed how well these needs must be complied with, i.e., the assurance with which the security needs must be designed and implemented. In our case, we focused on the ten most important assets defined, which enabled us to identify about twenty or so feared events, with half a dozen impacts each. For example, *propulsion management* has two security needs: availability "A" and integrity "I". By adding the negation symbol, we derived two feared events: *unavailability of propulsion management* "¬A" and *alteration of propulsion management* "¬I". Thereafter, we have identified that the feared event affecting the availability of *propulsion management* leads to four impacts. The first one, which is an *accident*, was judged most critical. A second one, which is an *immobilization on the open sea for a duration above 6 h or during a manoeuvre*, was rated with an average severity. Additionally, two other impacts were rated low in terms of BIL: *immobilization on the open sea for a duration below 6 h* and *reputational damage*.

### 4.1.10. Hindsight on the Second Datasheet

The Obérisk datasheet allows for the documentation of up to 20 feared events, which should be sufficient for an iteration in most risk management studies. Again, this is key for rallying the stakeholders. If the identified security needs require the analysis of more than 20 feared events, a new instance of the datasheet can be created. It should never be required to go above 40 feared events. If such a condition occurs, the business assets are probably identified at the wrong abstraction level, and the list of security needs should be reconsidered.

### 4.1.11. System Architecture and Identification of Existing or Already Specified Security Measures

The third and last standard Obérisk datasheet for workshop n°1 (cf. Figure 6) is the continuation of the first sheet. It is again called *Framing and security baseline*. It is split in two sections, numbered four and five. Section four allows documenting the supporting assets, represented by light blue sticky notes, such as the business assets. The supporting assets are organized into three groups:

- At the top, the organizational assets, such as human resources and their structuring;
- In the middle, Information Technology (IT) or Operational Technology (OT) assets, covering hardware, software and networks;
- At the bottom are significant physical assets used but are not usually an integral part of the system-under-study; this includes premises where the system-under-study is located and large infrastructures used by the system-under-study, e.g., the World Wide Web.

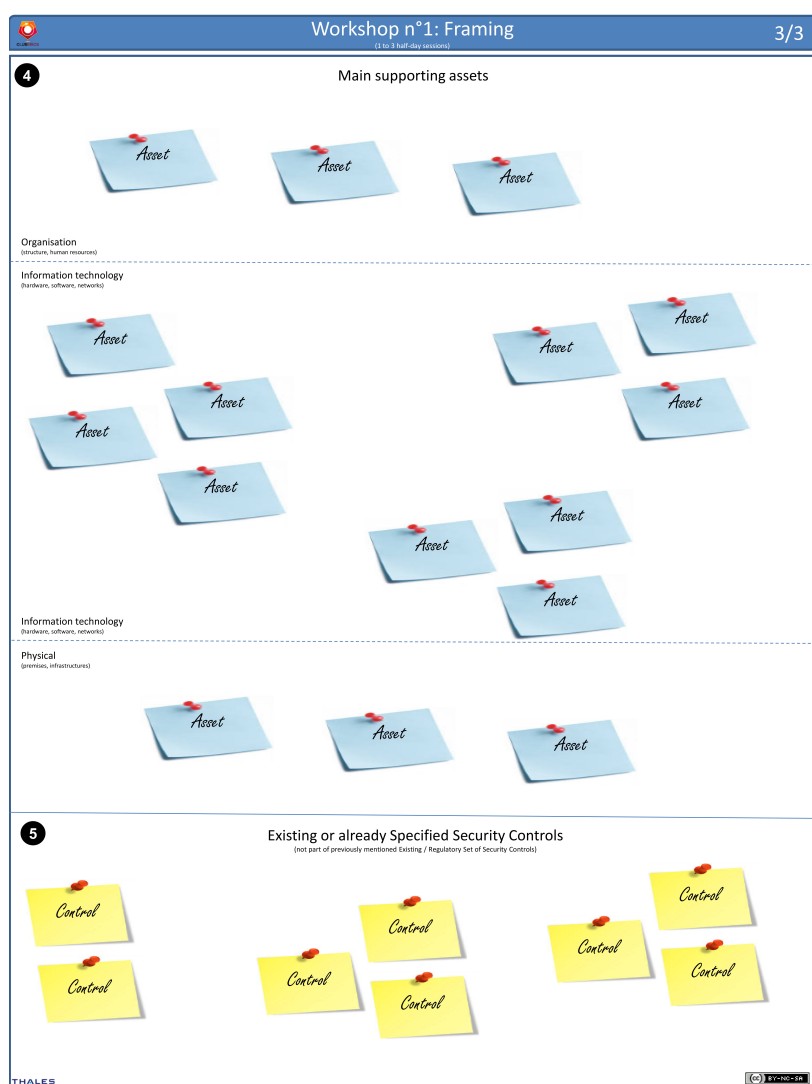

**Figure 6.** Obérisk datasheet—workshop n°1: framing and security baseline, part 3 of 3.

It is often practical for the IT/OT supporting assets to copy–paste a functional architecture schema on the Obérisk datasheet and annotate the architecture with sticky notes where and when required.

On the first Obérisk datasheet, we saw that Section 3 was dedicated to applicable regulation, policies and security standards. However, some security measures already

existing in the system-under-study or already specified for the system-under-study may have not directly derived from existing documents. Section five of the third standard Obérisk datasheet allows documenting the individual security measures that cannot be mapped to existing standards or policies.

Obérisk is designed for use in the early phases of a system's lifecycle, typically during the bid phase or the early design phase. The above datasheet should normally be sufficient to capture the essence of the system's architecture and provide a good overview of the existing or already specified security controls. However, if the architecture is complex and the work on security controls already well engaged, it may be relevant to provide more details. For these particularly complex cases, Obérisk offers to replace the above datasheet with two datasheets; thus, allowing for much more space for the supporting assets and for the security controls (cf. Figure 7).

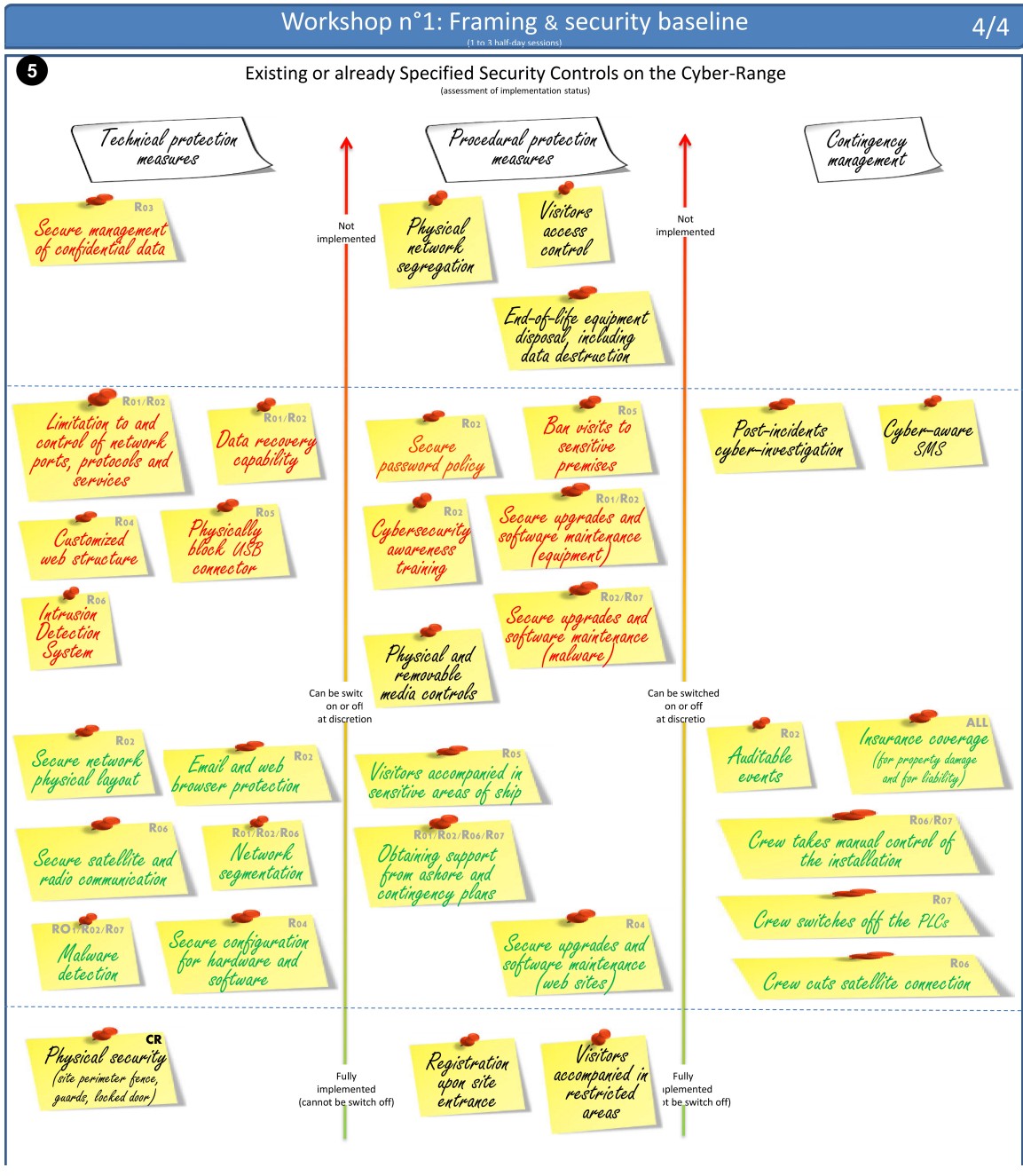

**Figure 7.** Naval use case—security baseline.

4.1.12. Illustration in a Naval Setting

Our naval use case was a study in which the architecture was already well defined, and the list of security controls already significant. We, therefore, opted to use Oberisk's two-datasheet alternative to have sufficient space. We will not discuss the architecture herein for confidentiality reasons, but the set of security controls is presented and discussed below (cf. Figure 7).

The synthesis of existing security controls was not easy to draw because it was initially unclear whether the synthesis should be representative of the cyber range, or a typical ferryboat or the best-of-class ferry. We finally decided to be representative of the cyber range, capturing:

- The unavoidable security controls, i.e., the controls related to the premises where the cyber range is installed.
- The security controls that can be switched on or off at will on the cyber range. Colour coding is used to specify if the security controls were considered as always active (green), sometimes active (black) or deactivated (red) in the attack scenarios. Thus, the implementation status presented in Figure 7 is not representative of a real situation on board. It is intended to be editable at will, in order to allow for the development of attack scenarios, which is one of the platform's key objectives.
- The security controls that may exist on certain ships but are not implemented in the scope of the cyber range.

Additionally, the security controls were split in three classes: technical protection measures, procedural protection measures and contingency management.

As stated previously, the platform had to respect and apply the European Union General Data Protection Regulation (EU GDPR), which led to several security measures both on a technical level, such as *email and web browser protection* or *secure management of confidential data*, and on a procedural level, such as *secure password policy*. The other measures were taken from several shipboard safety control guides.

Lessons learnt: This datasheet can be tedious to realize because it requires the inventory of all the security controls that the system must respect and their state of application. However, the result gives a clear view of the true security status of the system. This datasheet is used later during workshop n°4 (cf. Section 4.4.1), as a basis for the risk likelihood assessment, in order to see if existing security measures have a favourable impact on the risk scenarios and their participants.

*4.2. Workshop n°2: Risk Origins*

The second workshop of the EBIOS Risk Manager method is called *Risk Origins*. Its goals are to identify who or what may jeopardize the business/operational assets identified during the previous workshop and to what end. Its participants should be a top manager, a domain expert, the CISO and, if possible, a Threat Intelligence (TI) expert. The expected output is a prioritized map of the risk origins and their objectives. Workshop n°2 is supported by a single Obérisk datasheet, called *Identification of adverse objectives*. This sheet is split into two sections. The first section (cf. Figure 8) relates to Risk Origins (ROs), also known as risk sources, i.e., which are likely to try attacking the system-under-study. The second section (cf. Figure 9) relates to the Target Objectives (TO), i.e., what the attackers plan to achieve. A key workshop output is the relevant set of RO/TO pairs.

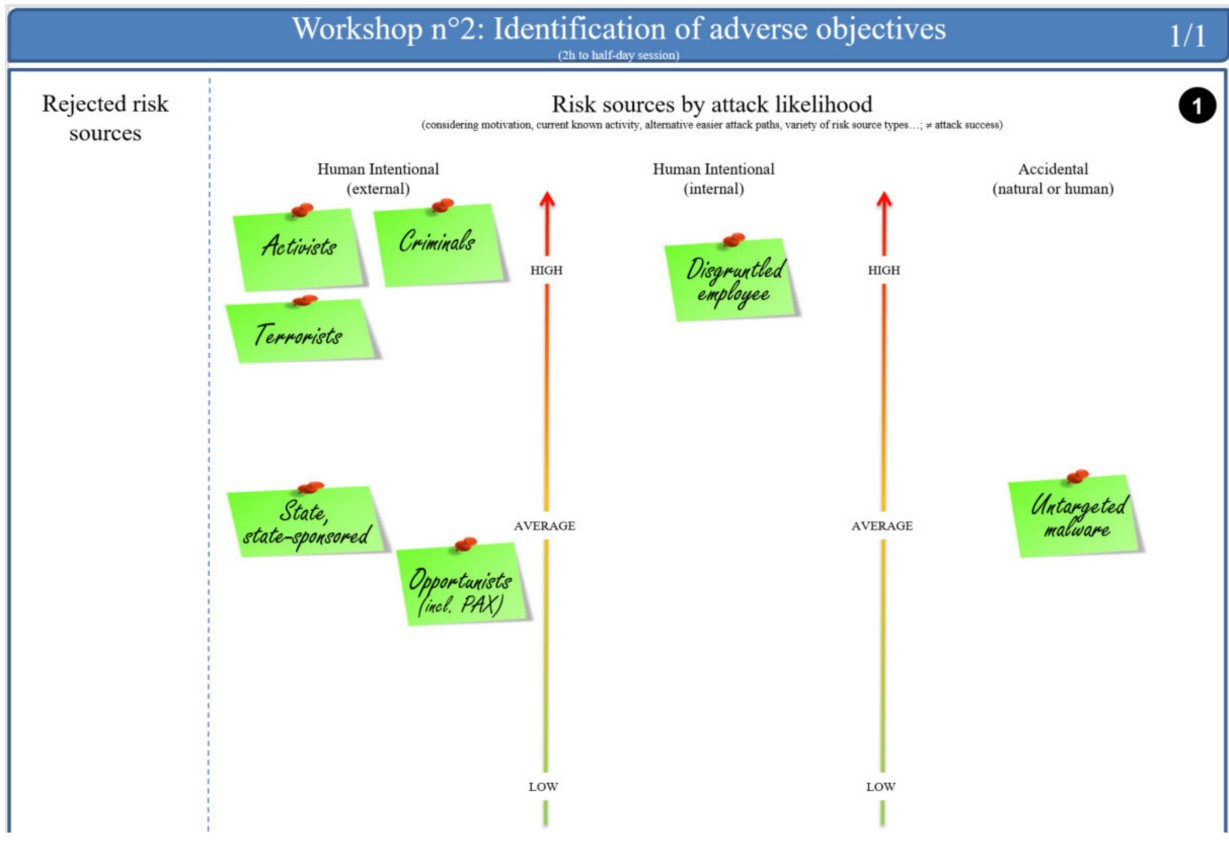

**Figure 8.** Naval use case—risk sources and their likelihood to try attacking.

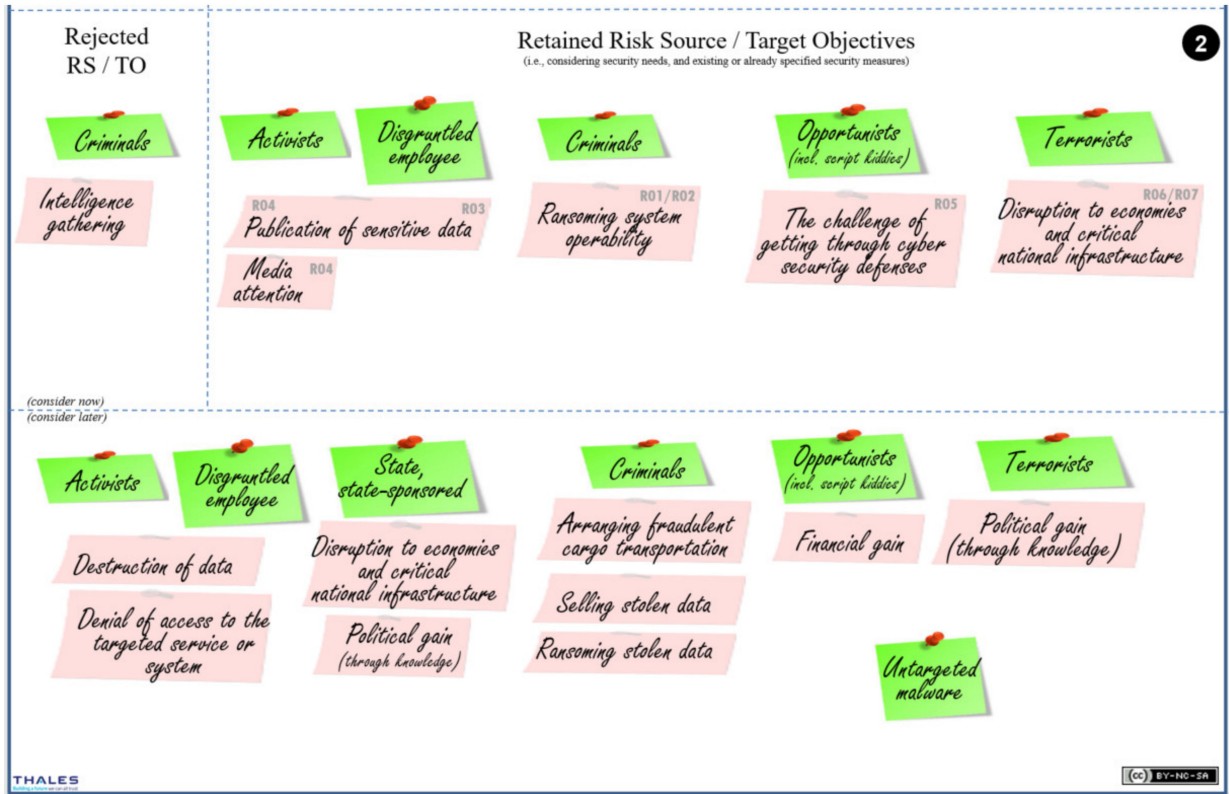

**Figure 9.** Obérisk datasheet—workshop n°2: identification of adverse objectives.

### 4.2.1. Identification and Assessment of the Risk Origins

In Section 1, the Obérisk layout offers three columns to host risk origins reified within green sticky notes. Each column allows for the capture of one type of risk origin:

- Column n°1: *human intentional (external)* risk origins; here, *external* means not belonging to the organization of the sponsor of the risk analysis;
- Column n°2: *human intentional (internal)* risk origins; here, *internal* means belonging to the organization of the sponsor;
- Column n°3: accidental (natural or human) risk origins.

Note: the EBIOS Risk Manager method does not usually consider accidental risk origins, based on the hypothesis that the security baseline defined during workshop n°1 is sufficient to deal with accidental risks. If the Obérisk datasheet offers space to host accidental risk origins, it is more related to human psychology than practical cyber security. Indeed, during a brainstorming session, the attendees will inevitably think and propose some accidental risk origins. Rebuffing a proposal may be experienced as stigmatizing: it may undermine the participants' enthusiasm and collaboration spirit. Having a column present on the datasheet will show that all ideas are considered, even if there is no follow-up.

On the Obérisk datasheet (cf. Figure 8), the risk origins are characterized by their likelihood of attacking, by positioning the sticky note along a vertical likelihood of attack axis. A position of the sticky note towards the bottom green part of the axis means that the risk origin is unlikely to try. A position of the sticky note towards the top red part of the axis means that the risk origin will certainly try. At this stage, no hypothesis is made on the potential success of the trial. Section 1 also offers a narrow column (on the extreme left) to list rejected risk origins. Indeed, it is essential to capture elements that were considered and wilfully rejected, by contrast to elements that were just forgotten during the analysis. This is particularly important for an audit, accreditation or certification.

### 4.2.2. Identification and Assessment of the Target Objectives

The second section of this datasheet (cf. Figure 9) relates to the risk origins' target objectives. First, the risk origins assessed as the most relevant in Section 1 are selected and copied in Section 2. Then, below each retained risk origin, target objectives are captured within pink sticky notes. They formed Risk Origin/Target Objective (RO/TO) pairs from which attack scenarios will stem in workshop n°3. At this stage, there may be many RO/TO pairs to consider. However, since the EBIOS Risk Manager method is not exhaustive, only the most relevant pairs must be selected for analysis. The Obérisk datasheet allows space for a reasonable amount of RO/TO pairs. When out of space, the risk analyst should consider postponing the analyses in order to focus on the most critical pairs. The bottom part of the sheet is a placeholder for RO/TO pairs stored for analysis during a future risk assessment iteration. Finally, the extreme left part of this section is designed to hold RO/TO pairs considered and wilfully rejected.

### 4.2.3. Illustration in a Naval Setting

Based on the expertise of the participants, the scientific literature and the available guidelines our study's most likely risk origins were defined as *activists, criminals, terrorists* and *disgruntled employees* (cf. Figure 8).

Then, we had to determine the objectives they could have or aim for. For example, we determined that a *criminal* could intend to perform the following actions: *ransoming system operability, arranging fraudulent cargo transportation, selling stolen data, ransoming stolen data, intelligence gathering*. After this first work, we classified the RO/TO couples according to their plausibility. We considered that the couple *criminal/ransoming system operability* should be treated immediately and that *arranging fraudulent cargo transportation, selling stolen data, ransoming stolen data* are to be kept, but treated later. The *criminal/intelligence gathering* couple has been discarded after discussing and placed in its respective column on the left.

Lessons learnt: the result of this work allows us to quickly see the sources of risks, their objectives and the most likely pairing between them.

*4.3. Workshop n°3: Strategic Scenarios*

The third EBIOS Risk Manager workshop describes high-level scenarios stating how the previously identified risk sources can attack the system-under-study. ANSSI asserts that a significant part of cybersecurity attacks does not target the system directly, but first targets the ecosystem, i.e., external actors interacting with the system-under-study, and then moves laterally to attack the system. Thus, before describing strategic scenarios, the EBIOS Risk Manager method mandates mapping the ecosystem and identifying critical participants, i.e., those most likely to be targeted by an attacker. The workshop participants should be a domain expert, an architect, the CISO, and, if possible, a cybersecurity expert. The expected outputs are a mapping of the ecosystem, a list of critical participants, a prioritized list of strategic scenarios and a proposal of complementary security controls applicable to the ecosystem. Workshop n°3 is supported by two standard Obérisk datasheet layouts, with one alternative layout for the second datasheet.

4.3.1. Identification and Assessment of the Ecosystem Participants

The first Obérisk datasheet for workshop n°3 is called *Risk evaluation at ecosystem-level*. The EBIOS Risk Manager method suggests assessing each participant in terms of:

- The system's exposure to the participant (see Equation (1));
- The cyber reliability of the participant (see Equation (2)).

Chapter 5 of the method's supplements guide [57] provides metrics for scoring each criterion and a mathematical formula to compute the criticality of each participant quantitatively (see Equations (1)–(3)).

$$\text{Exposure} = \text{Dependency} \times \text{Penetration} \tag{1}$$

$$\text{Cyber Reliability} = \text{Cyber Maturity} \times \text{Trust} \tag{2}$$

$$\text{Participant Criticality} = \text{Exposure} \div \text{Cyber Reliability} \tag{3}$$

By contrast, the approach with Obérisk is qualitative rather than quantitative. The actors interacting with the system are rated by positioning the participants in green sticky notes on a radar-like diagram (cf. Figure 10). The diagram comprehends two axes:

- The horizontal axis relates to the system's exposure to the participant, where exposure is defined as a combination of system dependency on the participant and the rights/privileges that the participant has over the system;
- The vertical axis relates to the participant's cyber reliability, where cyber reliability is defined as a combination of cyber maturity (i.e., trustworthiness) and trust.

The diagram is initially a bit difficult to read because both axes are bidirectional. Typically, low participant cyber reliability is expressed in the middle of the vertical axis (red part of the axis), whilst high participant cyber reliability can be expressed by locating the sticky note either at the top or at the bottom of the diagram (green parts of the axis). There is no semantic difference between locating a sticky note at the top or the bottom. Likewise, a high system exposure to the participant is expressed in the middle of the horizontal axis (red part of the axis), whilst low system exposure to the participant can be expressed by locating the sticky note either at the right or at the left of the diagram (green parts of the axis). There is no semantic difference between locating a sticky note at the right or at the left. With this convention, when the system-under-study is highly exposed to a participant and this participant is not cyber-reliable, then the participant's sticky note is located in the centre of the diagram. On the contrary, if the system-under-study is not exposed and/or if the participant is cyber-reliable, then the participant's sticky note is located on one side. This layout has two major advantages:

- When a participant is critically located at the centre of the diagram, they are also at the centre of attention; with normal axes, the critical participants would all be located in the upper right corner of the diagram;
- There is a lot of space on the sides of the diagram to group non-critical participants per type; for example, end-users can be located on the left, while maintenance operators can be located on the right, and internal staff can be located on the top, while external actors can be located at the bottom part of the diagram.

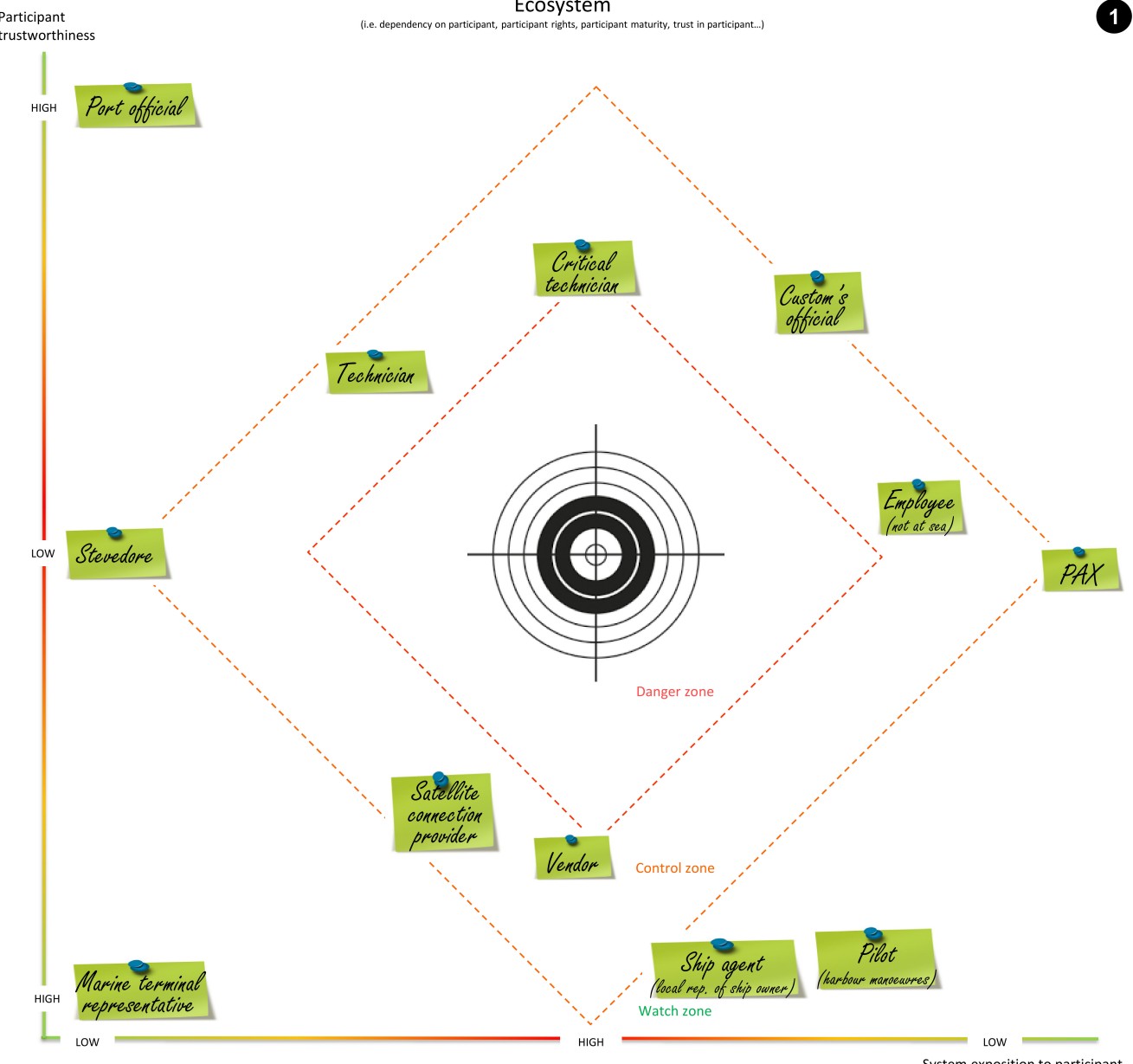

**Figure 10.** Naval use case—identification and assessment of the ecosystem participants.

The chart layout requires some oral explanations at the beginning of the workshop, but the return on the experience shows that it is easily accepted. This layout was recommended to us by the French cybersecurity agency (ANSSI) during a review of the very early drafts of the Obérisk datasheets. The layout has proved to be very efficient since then.

The radar-like diagram also displays two lozenges, delimitating three zones: the watch zone (in green), the control zone (in orange) and the danger zone (in red). The

choice of the lozenge shapes was driven by the French national cybersecurity agency's mathematical formula to compute each participant's criticality quantitatively [57] (see Equations (1)–(3)).The lozenges precisely reflect the numerical values of the mathematical formula without any cumbersome computations.

### 4.3.2. Illustration in a Naval Setting

In our naval context, we have identified 12 participants interacting with our system, as shown in Figure 10, each having different levels of cyber reliability and exposure. After determining their location, we could observe that the most critical participants are the *critical technicians* and the *vendors* because the system is highly exposed to their actions, and their cyber reliability is rated average. *Technicians*, *satellite connection providers*, and *employees* also need to be monitored because of their average cyber reliability and the average level of system exposure.

Lessons learnt: These results did not come as a surprise. People having direct access to the system represent, in most cases, a danger. In our case, any person external to the ship having to intervene on board to carry out maintenance, change a part, update software or carry out maintenance must be regarded as potentially dangerous. This level of danger must then be balanced with the person's cyber maturity. We can presume, for example, that the naval personnel will have greater integrity thanks to their training, awareness, and regulation, by comparison with a standard subcontracting company. This diagram allowed us to identify the stakeholders representing risk situations quickly. Therefore, it can be used as a communication tool with external parties.

### 4.3.3. Textual Definitions of Strategic Scenarios

The second standard Obérisk datasheet for workshop n°3 is called *Strategic scenarios*. It allows capturing textual expressions of strategic scenarios and rating them in terms of severity. The datasheet is split into three zones:

- Risks by severity (cf. Figure 11): the study of these risks will continue in the following workshops.
- Postponed risks: These risks are considered less important; the study of these risks is postponed until a future iteration of the risk assessment; since the next iteration may be run months later, these risks are register here in order to ensure they will not be forgotten.
- Rejected risks: These risks are considered irrelevant in the scope of this study and are wilfully rejected; usually a justification is required here. The rejection justification can be captured in the slide notes. These risks are not to be confused with accepted risks. Risk acceptance is part of risk treatment, and will be handled during workshop n°5 (cf. Section 4.5). As previously mentioned, it is always important to capture elements that were considered and wilfully rejected, by contrast to elements that were just forgotten during the analysis.

The risks by severity zone is split into four columns, one for each of the confidentiality, integrity and availability security criteria, plus one for privacy concerns. Vertical severity axes separate the columns. The general idea is to position the strategic scenario descriptions, on pink sticky notes, along the axes, depending on their level of severity and the main security criterion at play. A strategic scenario with a high severity will be located towards the top (red part), whilst a strategic scenario with a low severity will be located towards the bottom (green part).

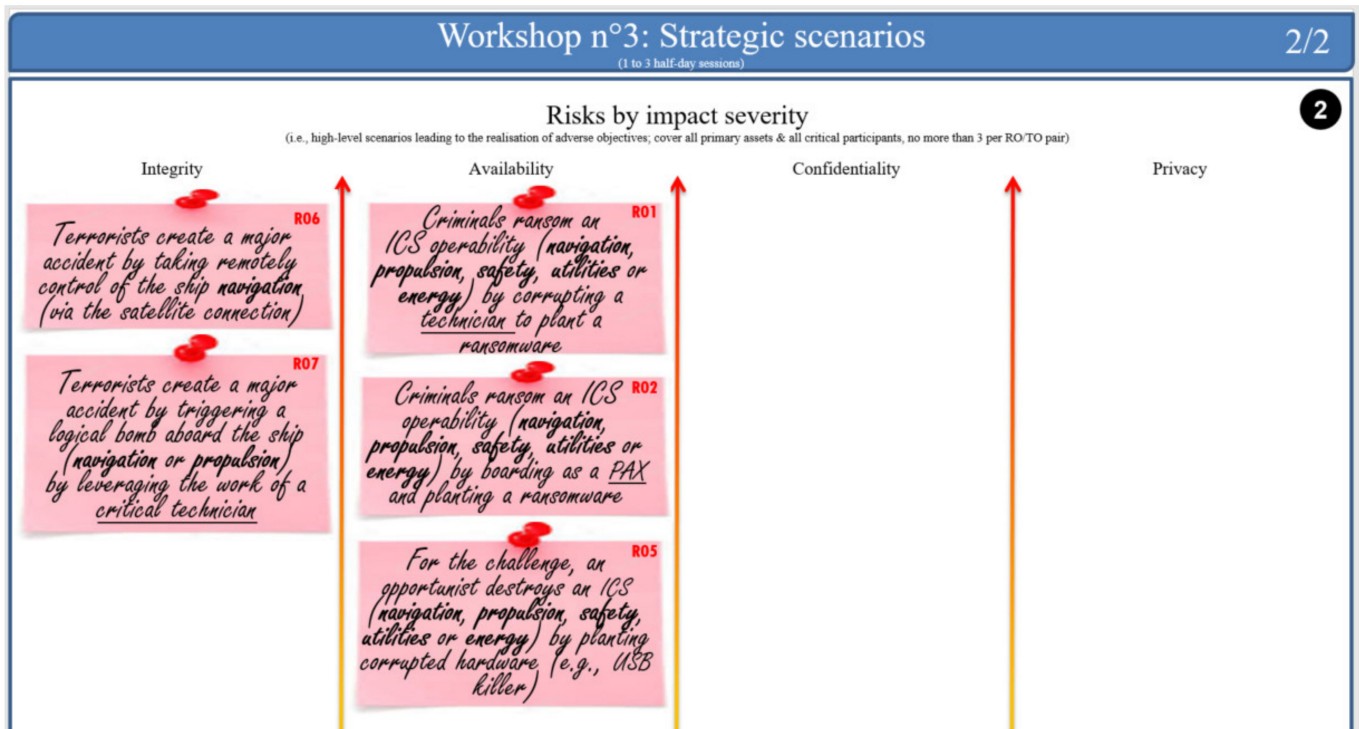

**Figure 11.** Naval use case—strategic scenarios (textual format).

Per default, some dummy strategic scenarios are already located on the datasheet waiting to be filled in during the brainstorming sessions. A text is proposed in the dummy strategic scenarios to recommend the writing syntax of well-formed strategic scenarios: <risk origin> targets <business asset> directly | via <participant> triggering <feared event>.

As mentioned previously, the EBIOS Risk Manager method is not an exhaustive method. Thus, only the most relevant strategic scenarios should be captured here. If there is no space left on the datasheet, one should consider postponing strategic scenarios until a further iteration.

It is a good practice to number all strategic scenarios with a risk number. This is shown by the *Rxx* labels on the top right part of the sticky notes, where xx represents a two-digit number. Moreover, it is a good practice to colour-code this label based on the strategic scenario's severity. Thus, when the label of a strategic scenario is copy–pasted onto the operational scenario datasheet (cf. Section 4.4), its severity will remain obvious. Colour coding is also very useful for postponed or rejected risks since the datasheet does not offer a severity axis for these strategic scenarios.

### 4.3.4. Illustration in a Naval Setting

Following this template, we have developed a variety of scenarios that stem from the RO/TO couples identified during workshop n°2. For example, *terrorists target the ship navigation*, via *the satellite connection, triggering a major accident*. The severity of this strategic scenario is derived from the business impact level of the related feared event, as seen during workshop n°1.

Lessons learnt: Each strategic scenario description must make adequate references to the feared event at play and the risk origin involved. Again, this can serve as communication support with external stakeholders or decision makers, allowing them to easily understand the most plausible and dangerous scenarios. All the key elements are presented in the scenario: the risk source, the possible participants used as attack vectors, the primary assets targeted and the method used to achieve the feared event.

### 4.3.5. Alternative Approach to Strategic Scenario Modelling

Obérisk proposes a textual representation of strategic scenarios and simplifies the EBIOS Risk Manager method by dismissing the concept of the attack route. The EBIOS Risk Manager method normally proposes a graphical representation of strategic scenarios, in which a strategic scenario may be broken down into multiple attack routes. For compliance reasons, Obérisk also proposes a graphical representation, as an alternative datasheet template (cf. Figure 12). This layout supports the concept of attack routes. It covers the three types of attack routes:

- The direct attack of a business asset located within the system-under-study. This type of route is noted *R0y*.
- The indirect attack of a business asset located within the system-under-study; here, the risk origin uses a participant as an attack vector. This type of route is noted *R0z*.
- The indirect attack of a business asset located within a participant's organization or information system (original asset or a copy of the asset). This type of route is noted *R0x*.

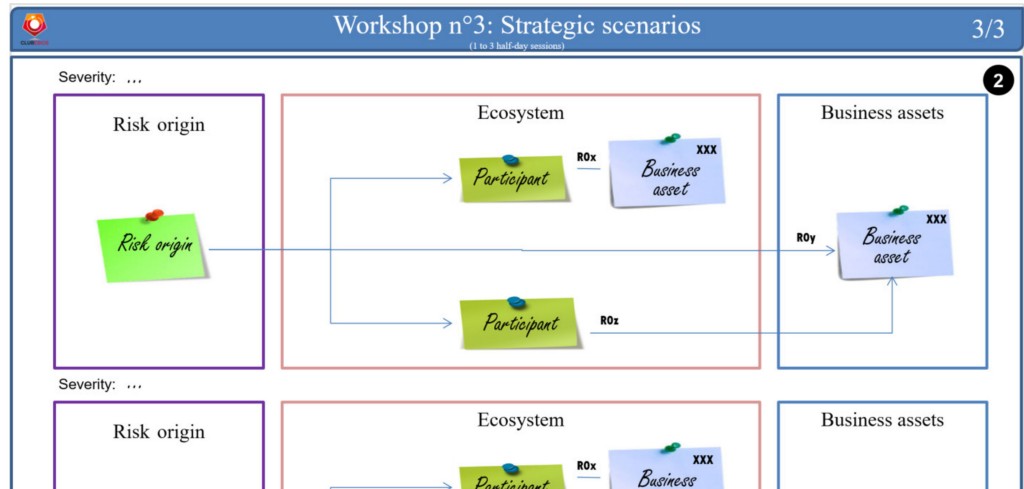

**Figure 12.** Obérisk datasheet—workshop n°3: strategic scenarios (graphical format).

With this graphical representation, the severity of each strategic scenario is captured textually just above the strategic scenario. This alternative datasheet is only recommended when there is only a small number of strategic scenarios. Otherwise, it will be necessary to multiply the occurrences of this datasheet, which will be detrimental to gaining an overview of the risks.

### 4.4. Workshop n°4: Operational Scenarios

The goals of the fourth EBIOS Risk Manager workshop are to describe how the strategic scenarios can be implemented using an ad hoc version of the Lockheed Martin Cyber Kill Chain® [58], and to assess their likelihood of success. Thus, this workshop requires a good knowledge of the supporting assets (as captured during workshop n°1) and their vulnerabilities. The resulting attack scenario is called an operational scenario in the EBIOS Risk Manager method. The workshop participants should be the CISO, the CIO and, preferably, a cybersecurity expert. The expected output is a list of operational scenarios with their likelihood of success. Workshop n°4 is supported by two standard Obérisk datasheet layouts and two alternative layouts.

### 4.4.1. Cyber Kill Chain Definitions

The first Obérisk datasheet for workshop n°4 is called *Risk evaluation at technical-level* (cf. Figure 13). This datasheet provides the canvas for an operational scenario description

based on a Cyber Kill Chain, but with some significant updates compared to the format proposed in the EBIOS Risk Manager guide [9]. The Cyber Kill Chain proposed by Obérisk is represented with the classical chevron symbol for processes [59]. It comprises five phases: (i) *external recognition*; (ii) *intrusion*; (iii) *internal recognition*; (iv) *lateral move*; (v) *exploitation*. Some backward arrows recall that, during an attack, the attacker may renew some previous phases, generally on another part of the system-under-study. The number and names of phases modelled in the Cyber Kill Chain are arbitrary, so Obérisk users may want to tune them before running a risk assessment.

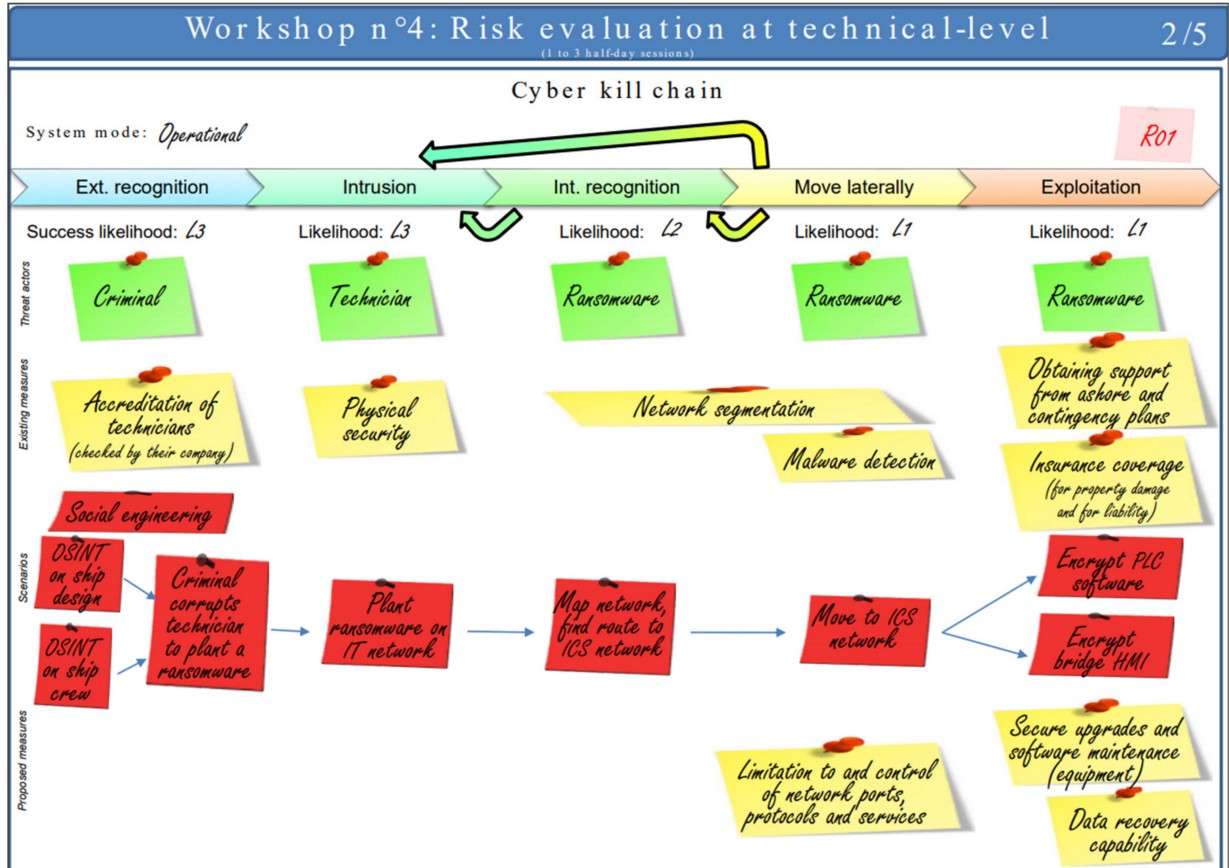

**Figure 13.** Naval use case—risk evaluation at technical-level (standard).

On the top-right of the operational scenario, a pink sticky note refers to the strategic scenario from which this operational scenario is derived. A small space can capture the system mode in which this scenario is run on the top-left of the operational scenario. Typical system modes include development, deployment, operation, maintenance, dismantlement, etc. The highlighting of the system mode is an addition of the Obérisk approach compared to the EBIOS Risk Manager method. Below the Cyber Kill Chain processes, space is allowed for five types of information, from top to bottom:

- Success likelihood for each phase of the Cyber Kill Chain;
- Threat actors, using a green sticky note, such as in workshop n°2;
- Existing security measures, using yellow sticky notes, such as in workshops n°1 and 3;
- Elementary attack actions, using red sticky notes;
- Newly proposed security measures, again using yellow sticky notes.

The likelihood considered here is solely the likelihood of success. The likelihood of occurrence, i.e., the fact that the risk origin tries to attack, has already been considered in workshop n°2. The overall risk likelihood will need to combine the likelihood of occurrence and the likelihood of success. The likelihood of success may be expressed qualitatively

using a scale matrix or quantitatively, e.g., using a percentage. In either case, return on experience shows that a cumulative assignment of the likelihood of success as the scenario progresses is very practical. Indeed, in that case, the value below the exploitation phase corresponds to the overall scenario likelihood of success. On the contrary, if the likelihood of success of each phase is set independently from the other phases, then it is left to the reader to compute the overall scenario likelihood of success.

Similarly to the HMG IA Standard 1 & 2 [60], Obérisk distinguishes the threat origin (also known as threat source in the HMG IA Standard), i.e., a person or organization that desires to breach security and ultimately will benefit from it, from the threat actor, i.e., a person who performs an attack or, in the case of accidents, will cause the accident. The datasheet allows for the specification of one or more threat actors at each phase of the Cyber Kill Chain. This is an addition of Obérisk compared to the standard EBIOS Risk Manager method. Indeed, we considered that the knowledge of the risk actor is essential when assessing the likelihood of success of an attack.

Likewise, the modelling of the existing security measures on the Cyber Kill Chain is an addition of Obérisk compared to the standard EBIOS Risk Manager method. Again, this extension was proposed because it was felt that considering the existing security measures, it is very important when assessing the likelihood of success of an attack. To remain legible, only the security measures with a significant impact on the likelihood of success are captured on the diagram.

Finally, the modelling of the newly proposed security measures is again an addition to the Obérisk approach. In the standard EBIOS Risk Manager method, the risk treatment work is performed in workshop n°5. We considered that establishing a Cyber Kill Chain requires a high level of concentration by the workshop participants. When establishing the Cyber Kill Chain as part of workshop n°4, proposing some security measures to improve the situation is an easy and natural task. Doing the same work as part of workshop n°5, possibly days or weeks later, will require much more time and effort. However, it is to be noted that the proposal of some additional security measures with Obérisk in workshop n°4 should not be confused with risk treatment. Even with Obérisk, risk treatment remains a workshop n°5 activity, because the risk treatment requires a global view over all risks. Risk treatment should not be handled locally at the level of a single operational scenario. The objective behind the listing of proposed security measures during workshop n°4 is to ease selecting the appropriate security measures during the risk treatment.

The standard operational scenario datasheet allows for describing two scenarios on the same page. An alternative operational scenario datasheet allows for the documentation of a particularly complex attack scenario. The layout is similar, but more space is allocated to capture the (existing/proposed) security controls and the elementary actions.

### 4.4.2. Risk Registry

The second standard Obérisk datasheet for workshop n°4 is the *Risk registry*. This is the first of the two Obérisk datasheets that are essentially textual, rather than graphical. The second one will be the overall study conclusions. The risk registry allows capturing the risk descriptions, the risk owners and some comments. The standard Obérisk risk registry datasheet allows for the description of up to 15 risks on the same page. An alternative risk registry datasheet allows for the documentation of fewer risks. However, it offers a qualitative risk likelihood scale, which can also be used to assess the likelihood of the operational scenarios (cf. Figure 14).

Risk likelihood scale

| SCALE | DEFINITION |
|---|---|
| **L4 – CERTAIN OR ALREADY OCCURRED** | The risk origin will certainly reach its target objective by one of the considered methods of attack OR such a scenario has already occurred within the organisation (incident history). |
| **L3 – VERY LIKELY** | The risk origin will most probably reach its target objective by one of the considered methods of attack. |
| **L2 – LIKELY** | The risk origin is able to reach its target objective by one of the considered methods of attack. |
| **L1 – RATHER UNLIKELY** | The risk origin has little chance of reaching its objective by one of the considered methods of attack. |

**Figure 14.** Obérisk datasheet—workshop n°4: qualitative risk likelihood scale.

### 4.4.3. Illustration in a Naval Setting

Figure 13 illustrates one of the operational scenarios identified during the naval use case analysis. In this scenario, during the *External reconnaissance* phase, we assumed that the *criminal* will perform *Open-Source INTelligence (OSINT) on the ship's design and on its crew*, as well as *social engineering*, to find a way *to coerce or corrupt the technician to plant malware* into our system. This step was our entry point to move to the *Intrusion* phase represented by the action of the *technician* who *plants the ransomware in the IT network*. The *ransomware* then *maps the network* (*Internal recognition* phase), to *find the route to the Industrial Control Systems (ICS) network*. Once the route is identified, the *ransomware moves laterally to the ICS network* to start the *Exploitation* phase, for example, *Encrypt Programmable Logic Controller (PLC) software* or *Encrypt bridge Human–Machine Interface (HMI)*.

For each phase of the Cyber Kill Chain, we have assigned a level of likelihood based on the proposed qualitative risk likelihood scale (cf. Figure 14). Using this scale, we determined that:

- The feasibility of steps one and two was *very likely* (L3), mainly due to the human factor, which is very difficult to control;
- The cumulative feasibility of step three drops to *likely* (L2), because it concerns our system network exclusively, and we have a certain degree of control over it;
- The cumulative feasibility of step four drops again to *rather unlikely* (L1) given the security measures already implemented to prevent lateral movement.
- Seven operational scenarios were described in this fashion for our naval use case.

In Figure 15, we can see four risks for the naval use case, including risk R01 presented above. The R01 risk relates to the introduction of ransomware, leading to the loss of some ICS operations necessary to manage propulsion, energy and other critical functions. Referring to the feared event defined in workshop n°1, we concluded that this scenario had a *high severity*. However, we have seen during the development of its operational scenario in workshop n°4 that the likelihood of realizing each of these steps was *rather unlikely*.

If we take R03 as a second example, we see that this scenario describes a disgruntled employee accessing and disclosing operational data to damage the organization's reputation. In this case, we still referred to workshop n°1 and saw that the damage to the company reputation is considered less severe; however, as the threat is internal, the Cyber Kill Chain to achieve it is quite simple, leading to a high probability of realization if no measures are applied.

**Figure 15.** Naval use case—risk registry.

### 4.5. Workshop n°5: Risk Treatment

The objectives of workshop n°5, called *Risk Treatment*, are to synthesize the inherent/initial risks, define a risk treatment strategy, derive the corresponding security measures, integrating them in a continuous improvement plan and assess/manage the residual risks. The workshop participants should be, as for workshop n°1, a top manager, a domain expert, the CISO and the CIO. Three standard Obérisk datasheet layouts support workshop n°5.

#### 4.5.1. Risk Syntheses

The first Obérisk datasheet for workshop n°5 is dedicated to risk syntheses. It is split into two parts. At the top, the synthesis of inherent/initial risks, i.e., the risks before treatment (cf. Figure 16). At the bottom, the synthesis of residual risks, i.e., the risks after treatment (cf. Figure 17). Both syntheses are somehow classical in that they allow positioning the risks, reified as red sticky notes, based on their likelihood and severity. However, the Obérisk datasheet provides some filigreed hints for cybersecurity neophytes.

In the synthesis of inherent/initial risks, if a risk is positioned in the upper right corner, the filigreed hints recommend *risk modification* to diminish the risk likelihood or *risk-sharing* to diminish the risk severity. If a risk is positioned in the lower right corner, the filigreed hint recommends only *risk modification* to diminish the risk likelihood. If a risk is positioned in the upper left corner, the filigreed hints recommend *risk-sharing* to diminish the risk severity or *risk avoidance*, i.e., avoid the activity or condition that gives rise to the risk. Finally, if a risk is positioned in the lower-left corner, the filigreed hint recommends *risk retention/acceptance*. These are only recommendations and other risk treatment decisions may be taken.

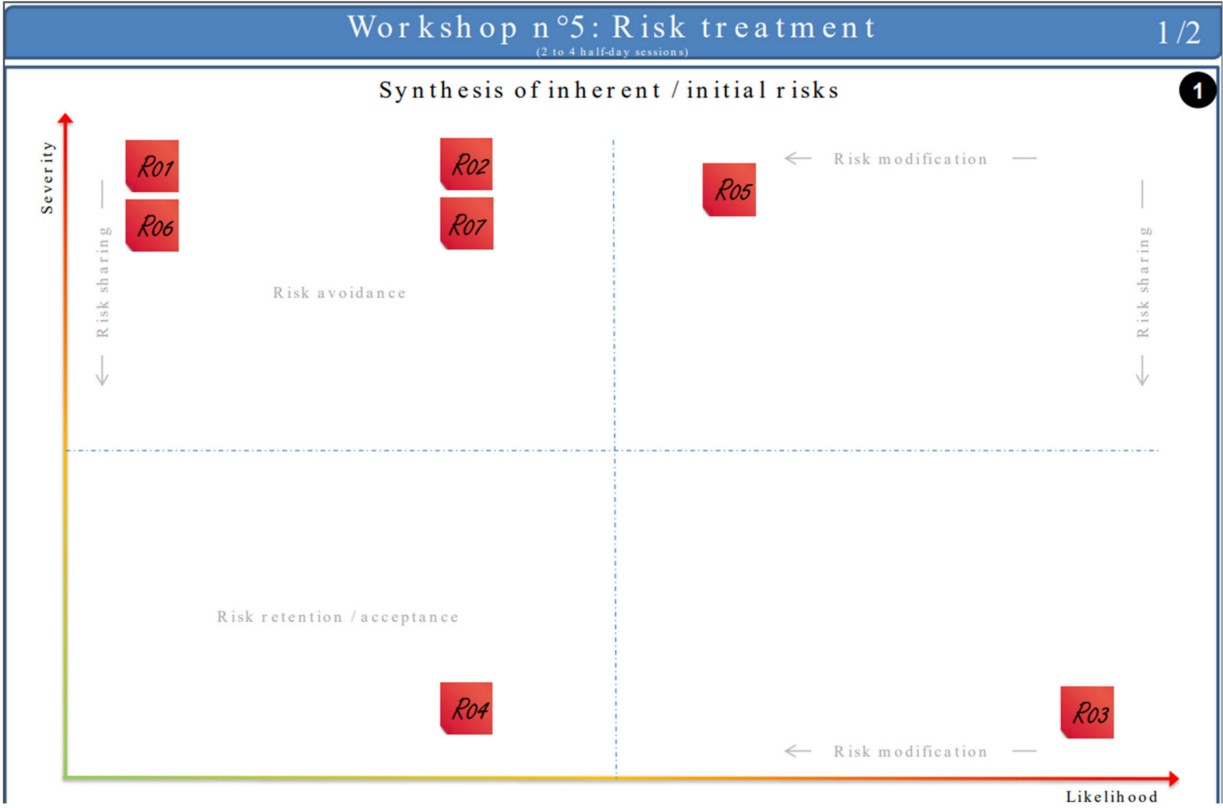

**Figure 16.** Naval use case—inherent risk synthesis.

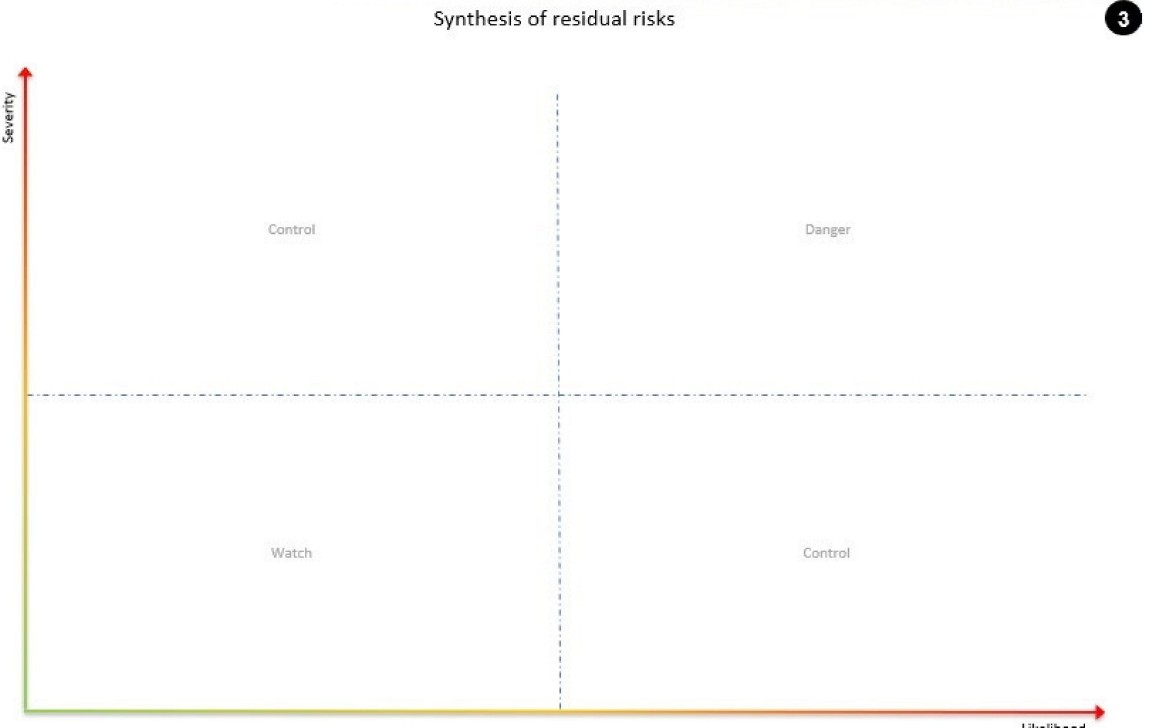

**Figure 17.** Obérisk datasheet—workshop n°5: synthesis of residual risks (template).

In the synthesis of residual risks, the filigreed text qualifies the risk level. If a residual risk is positioned in the upper right corner, the filigreed text indicates a *danger*. If a residual

risk is positioned in the lower right corner or in the upper left corner, the filigreed text indicates that the risk should be kept under *control*. Finally, if a residual risk is positioned in the lower-left corner, the filigreed text states *watch*, thus recalling that it is useful to review the risk from time to time. Indeed, it is always good to remind the stakeholders that accepting risk is not synonymous with doing nothing. One should typically consider time-bound security waivers to be documented by the product owner and validated by the cybersecurity authority.

### 4.5.2. Illustration in a Naval Setting

Figure 16 shows our seven risk scenarios presented in workshop n°4. Risk scenario R01 is unsurprisingly at the top left of our axis, along with the other risk scenarios (i.e., R02, R06 and R07) that also affect critical functions. The very likely R05 is significantly more to the right: it stands out as the most critical risk in our study. Finally, we placed R03 at the bottom right because it is certain to happen and has a very low severity.

The above representation of the risks (cf. Figure 16) is, in a way, the most interesting for the decision makers, coupled with the textual representation of the strategic scenarios (cf. Figure 11). They allow having a concise overview of the scenarios and their processing. Thanks to the workshops previously carried out, and the central thread between them, the work is synthesized and can be used to present and support the remarks of the security teams at the time of the decision of the security measures to be set up.

The objective of our risk assessment study was to identify the vulnerabilities and threats brought by the environment, architecture and naval system, in order to train the captain and crew with our cyber range on practical cases of cyber-attacks, which are not or cannot be solved easily by additional technical security measures. Thus, the naval case study illustration stops here, as we do not require running the next steps of the risk treatment workshop.

### 4.5.3. Risk Treatment

The Obérisk risk syntheses datasheet shows the two syntheses on the same page to quickly compare the risk level before and after treatment. It should, however, be noted that these diagrams are respectively numbered one and three (in black circles on the right side of the page). Step n°2 is the risk treatment itself. The risk treatment process is covered by an eponym Obérisk datasheet, which is split into four columns (cf. Figure 18). On this datasheet, risks are once again reified as red sticky notes. The first column is a placeholder for the risks that are formally retained/accepted. The second column relates to risk modification. This column is larger because this treatment option consists of introducing, removing, or altering security controls. The security controls are reified using yellow sticky notes, similar to workshops n°1, 3, and 4. The third column relates to risk avoidance. A light-blue sticky note allows for capturing the change of activity or condition that gives rise to the risk, possibly proposing an alternative activity or condition. Finally, the fourth column relates to risk sharing. A yellow sticky note is used once again to explain how the risk is shared with another party that can most effectively manage the particular risk. This is not really a security control, but the concepts were thought to be sufficiently close not to introduce any confusion.

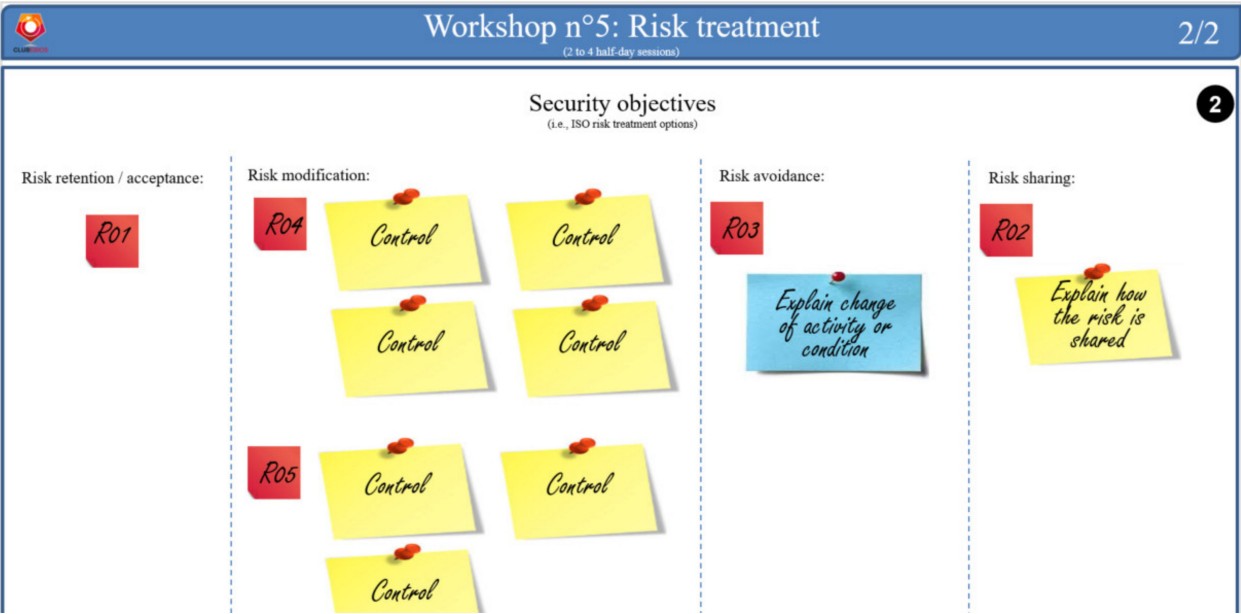

**Figure 18.** Obérisk datasheet—workshop n°5: risk treatment action plan (template).

### 4.5.4. Study Conclusions

The last Obérisk datasheet is called "Conclusions". This is the second of the two Obérisk datasheets that are essentially textual rather than graphical. The first one is the risk registry. This datasheet essentially offers free text to write an executive summary of the analysis. The datasheet also allows listing all the study contributors and their affiliation.

Overall, the complete list of datasheets represents 14 to 17 pages, which is an incredibly compact format for a full-blown risk assessment and risk treatment study.

## 5. Discussion

### 5.1. The Naval Use Case End-User Viewpoint

The first workshop of the naval platform risk analysis initially took place in a face-to-face meeting, with the "A0-format posters" support previously presented. Beyond the facilitator, the quite large gathering included four active participants to the risk assessment, and an audience of peers, including many PhD students, who came to see how this new risk assessment approach worked. The active participants were equipped with sticky notes and pens. They brainstormed under the supervision of the facilitator. Ideas were proposed, discussed amongst them and occasionally validated by peers in the audience. Once agreed, the text was written on a sticky note and placed by the facilitator at the decided location on the poster.

The evolution of the COVID crisis and the government's directives pushed us, thereafter, to carry out the remaining workshops in remote sessions. In this context, the facilitator shared his screen, with the Microsoft PowerPoint displayed. He remained in charge of the discussion and, after exchanging with the participants, he filled in and positioned the sticky notes himself.

These two approaches were found to have their advantages and disadvantages. The face-to-face approach allowed the teams to exchange more freely with each other, ask more questions and refine the definition and placement of elements. Body language also plays a more important role in the exchanges when the participants are physically present, especially when there are hierarchical relationships between attendants. The facilitator must ensure that the discussions remained within the scope of the study to avoid wasting time and loosing focus. In the remote session approach, it was undeniable that the preparation and planning were greatly simplified. However, we found that the participants

were less inclined to exchange details or question each other. Therefore, the facilitator must fully assume his role and emphasize the follow-up and participation of the participants to complete the study.

This experience allowed us to validate the relevance of the elements of the approach, as well as their representations, to identify and estimate the risks of a complex and demanding system/project. This was performed through practical use by a large panel of user profiles.

In the first workshop, the teams involved in the study quickly identified the objectives they wanted to achieve and the main tasks the system should provide. As the platform reproduced the operational conditions of an active ship, the participants were able to refer to existing cases and certification documents to identify and define the glossary, asset owners, business assets, and regulations. It should be noted that the validation and comments of the team members greatly assisted in this effort. Subsequently, the relative "rating/classification" of business assets without reference to probability, severity and criticality matrices were easier to achieve by business experts than by security experts.

However, summarising a feature with a simple sticky-note can lead people without a comprehensive knowledge of the field and, in our case, the security experts, to forget the subtleties and discussions that led to their reification as a stick-note at a specific spot on a poster. This effect became more pronounced as the study progressed. This problem could be solved either by making a voice recording of the exchanges (possibly with a speech-to-text conversion), or by writing down the justifications in the slide notes not included in the final report.

Concerning the feared events, the business experts greatly helped identify, categorize and classify the different impacts by giving real case examples. Unsurprisingly, without the support of a matrix to refer to, the security experts found it more challenging to maintain an overview and comparison. However, their questions and viewpoints exchanges with the business experts brought out the best in the latter.

For the second workshop, the participants used an existing standard threat baseline and then decided to short-list the threats on a case-by-case basis. Here, again, business expertise played a major role in the categorization and estimation of threats. Security experts were able to bring their knowledge on the attacker point of view, and thereof help define the target objectives of the attackers.

In the third workshop, business experts almost exclusively carried out the risk evaluation at the ecosystem level. The nature of the knowledge required focuses on the estimation of the cyber reliability of the actors interacting with the system and system exposure to these actors. By contrast, the second part of the third workshop, related to strategic scenarios and their estimation in terms of severity, was carried out collectively.

Workshops four and five were almost exclusively carried out by the security experts, while the business experts participated through their feedback on the situations they had encountered and managed; thus, improving the relevance of the scenarios.

Overall, through this experience, we saw that the business experts significantly contributed to the first workshops, whilst the security experts had more difficulty performing without their support. As the study progressed and security aspects came into play, the business participants took a back seat, serving as a knowledge base for the security experts.

The Obérisk tooled-up approach is complete and fully compliant with the EBIOS Risk Manager method. However, capturing risk information on slides has its limits. Obérisk cannot scale such as commercial tools (e.g., [32–34]). For example, Obérisk does not provide access to large knowledge bases, allowing easy access to catalogues of threats, vulnerabilities or security measures present in standards. Still, it has the merit of being quick to learn and intuitive in its implementation and execution. Similar to any agile method, success greatly depends on the quality of the facilitator and the participants, whose exchanges were simplified, thanks to the proposed ergonomics. In short, Obérisk allows a quick and relevant first analysis.

*5.2. Further Work*

Obérisk is currently focused on managing cybersecurity risks according to the EBIOS Risk Manager method. Some extensions of Obérisk are being considered as part of the PRAETORIAN project [61], in particular the capability to cope with physical threats, and other types of risks, in particular safety-related risks. This should hopefully trigger only minor adaptations.

We do not plan to extend Obérisk in order to comply with other risk management methods. If third parties are interested to do so, it is worthwhile stating that we believe that the core principles of Obérisk can easily be reproduced to handle any qualitative non-exhaustive approach to risk management. By contrast, an Obérisk-like approach to exhaustive methods, such as EBIOS-2010 [62], or quantitative risk management methods, such as FAIR [63], may be difficult or inappropriate.

**Author Contributions:** Conceptualization, S.P.; Funding acquisition, S.P.; Investigation, S.P.; Methodology, S.P.; Resources, S.P. and E.G.; Supervision, E.G.; Validation, S.P. and D.N.; Writing—Original draft, S.P. and D.N.; Writing—Review and editing, S.P. and D.N. All authors have read and agreed to the published version of the manuscript.

**Funding:** This research and its APC were co-funded by the Horizon 2020 Framework Programme FORESIGHT (H2020-SU-DS-2018)–Grant Agreement 833673, and Thales Research and Technology.

**Informed Consent Statement:** Not applicable.

**Data Availability Statement:** The templates and examples presented in this study are openly available at https://club-ebios.org/site/en/obeya-like-risk-management-approach/ (accessed on 25 August 2021).

**Acknowledgments:** We hereby acknowledge ANSSI and the College of Practitioners of Club EBIOS for their valuable comments on the method and material.

**Conflicts of Interest:** The authors declare no conflict of interest.

## Appendix A. Brief Overview of the EBIOS Risk Manager Method

In October 2018, the French National Agency for Cybersecurity (ANSSI) published a new version of the EBIOS risk management method called EBIOS Risk Manager [9], with a small revision in December 2018. It has been available in English since November 2019. The first version of the EBIOS method dates back to 1995. It was significantly updated in 2004 and 2010. EBIOS-2010 established itself as the main risk management method used in France.

The EBIOS Risk Manager method proposes five workshops called: (1) framing and security baseline; (2) risk origins; (3) strategic scenarios; (4) operational scenarios; (5) risk treatment. The method specifies the objectives of each workshop, the expected attendees, the expected outputs, and how to proceed.

Compared to EBIOS-2010, the new version of the method brought some significant changes, amongst which the following:

- It explicitly targets populations beyond the classical cybersecurity experts, including company directorates, risk managers, Chief Information Security Officers (CISOs), Chief Information Officers (CIOs) and business/operational experts, such as Architects (ARCs), Product Line Architects (PLAs), Design Authorities (DAs) and System Engineering Managers (SEMs).
- It mandates securing by conformity, prior to securing by scenario. Securing by conformity means that a Minimal Set of Security Controls (MSSC), based on best practices, is selected prior to risk identification. Then, risks are identified taking into account existing controls. ANSSI's hypothesis is that accidental events should normally be covered by the MSSCs, so that the analysis can focus solely on malevolence. In other words, only a few incident scenarios should be necessary, either to prove that the

solution is secure, or to highlight some rare holes in the system. This is very important in terms of scaling. If the study becomes big, you have something wrong!

- It is run as a set of short workshops (i.e., brainstorming sessions), with two to four participants per session.
- It is an agile approach, providing quick results for decision makers. Typically, it is possible to start outputting grosgrain risks after only three workshops. To go in depth, it is possible to iterate on the workshops. It is also recommended to iterate through operational and/or strategical cycles, to keep the system in secure conditions throughout its lifecycle. The operational cycle deals with fast changing facts, e.g., security measures, vulnerabilities. The strategic cycle deals with slower-changing facts, e.g., system missions, risk sources.
- It is configurable. More explicitly, it is not required to run all five workshops in sequence. The choice to run a workshop depends on the study objectives.
- Its scope is extended to include the ecosystem, also known as actors who interact with the system. ANSSI's assumption here is that many attacks do not directly target the system, but first target an actor in the ecosystem (e.g., a sub-contractor), then move laterally.

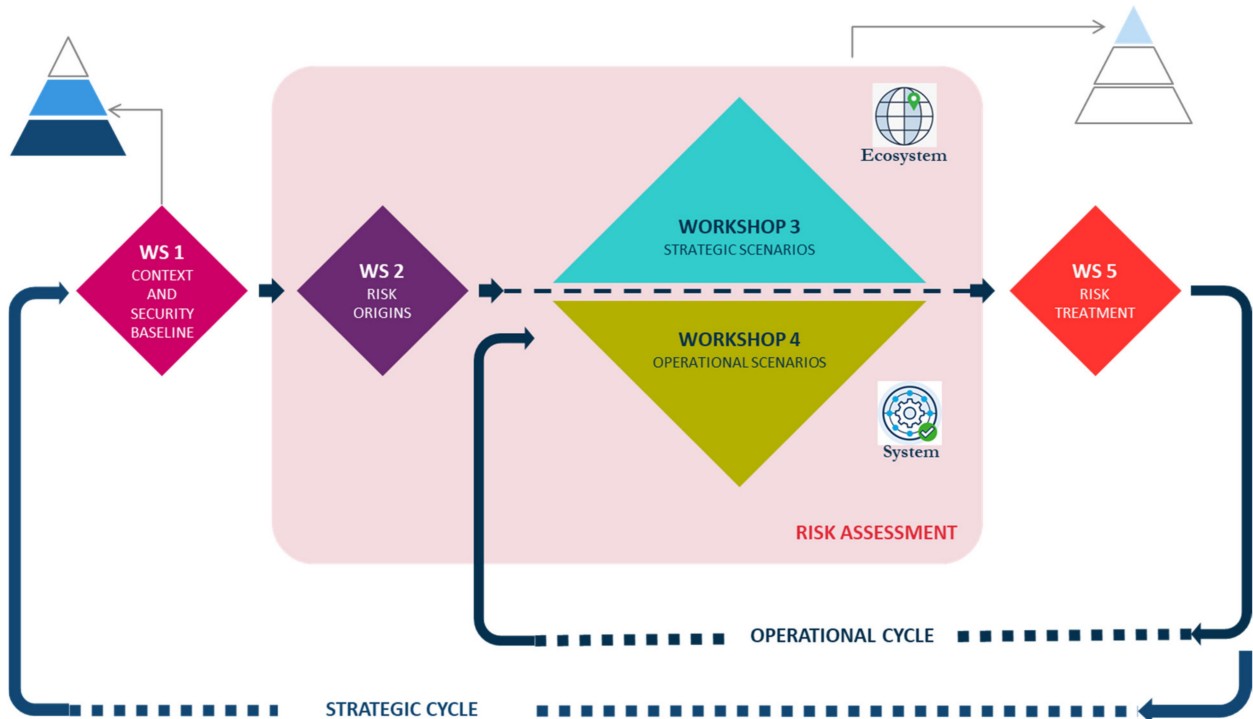

**Figure A1.** The five workshops of the EBIOS Risk Manager method.

The terminology used in this article is compliant with the EBIOS Risk Manager method. Readers unfamiliar with some terms are invited to refer to the glossary of the method's handbook [9].

**Appendix B. Obérisk Framework**

This appendix lists the standard Obérisk datasheets and the alternative datasheets for each workshop. The set comprises 18 different layouts, covering all the EBIOS Risk Manager method workshops. It includes 11 standard sheets, sufficient to perform a complete risk management study, and 7 alternative sheets. The alternative sheets were essentially designed to cope with scalability requirements, or ergonomic preferences.

**Table A1.** Obérisk framework.

| EBIOS-RM Workshop | Main Obérisk Datasheets | Alternative Obérisk Datasheets |
|---|---|---|
| 1 | Framing and security baseline (1/3) Business impact assessment (2/3) Framing and security baseline (3/3) | Framing (n°1) Framing and security baseline Framing (n°2) Security baseline |
| 2 | Identification of adverse objectives | |
| 3 | Risk evaluation at ecosystem-level Strategic scenarios (textual) | Strategic scenarios (graphical) |
| 4 | Risk evaluation at technical-level (2 scenarios per page) Risk registry | Risk evaluation at technical-level (1 scenario per page) Risk registry (with risk likelihood scale) |
| 5 | Risk syntheses Risk treatment Conclusions | |

In addition, the Obérisk framework comprises a sticky notes library. This library allows retrieving any sticky note useful for the datasheets. The sticky notes are organised per workshop (cf. Figure A2), so that the facilitator can easily retrieve all the relevant sticky notes with respect to his current concern.

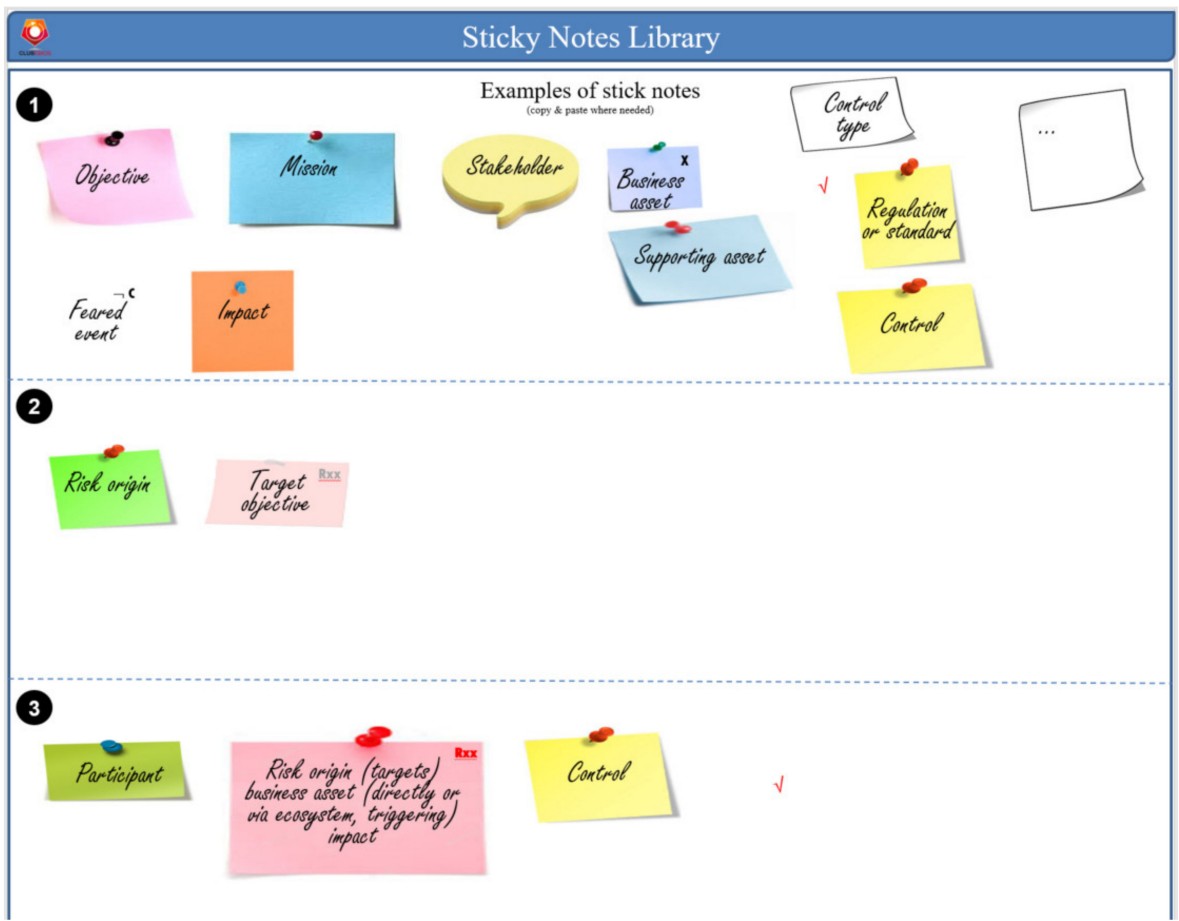

**Figure A2.** Stick notes library (extract).

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
