# Peer review of "Obérisk: Cybersecurity Requirements Elicitation through Agile Remote or Face-to-Face Risk Management Brainstorming Sessions"

_information, doi:10.3390/info12090349_

Round 1

Reviewer 1 Report

In this paper, the authors proposed a new, collaborative cyber risk management approach, Oberisk. The authors explained how this approach works and validated the Oberisk through a case study, which is a workshop of several participants.

In general, I see the promise of the cyber risk approach introduced in the paper. However, in order to advance the manuscript, the following issues must be addressed:

  1. The paper is more like a white paper than an academic journal paper. I appreciate the practicality of Oberisk. However, to be a journal paper, the novelty of Oberisk should be shown through a literature review, which is missing in the current version.
    1. Add a literature review section after the introduction. In this new section, academic, peer-reviewed sources should be analyzed to answer the following questions:
      • Why is this new method needed? - Knowledge gap: What others did and didn't do so far.
      • Why is this method novel?
      • Is it better than previous risk management approaches? If so, in which aspects?
    2. There is almost no peer-reviewed work cited in this paper. All of the references are either a webpage, standard, or report. The authors should cite peer-reviewed journal or conference papers in all sections of the paper - particularly in the new literature review section requested in (1.a). At least half of the references should be peer-reviewed papers after revision.
  2. I suggest adding another section, "Method", which is different than the current Section 2. Materials and Methods, to explain the method of this study not how Oberisk works. In this new Method section, the authors should explain the steps of this research - as far as I understand, it is composed of the following steps
    • Developing a novel cyber risk management method
    • Validating this method by running it on a particular case at a workshop
      • Data Collection: Who are the participants? detailed information about the number of participants and their profiles
      • Data Analysis: How is data collected and analyzed? There are lessons learnt sections in all steps of the workshop however, it is not clear that how the authors came up with these conclusions (e.g., a survey to participants after each step, expert opinion, etc.)?
  3. Rename current Section 2. Materials and Methods, to avoid confusion with the new Methods section. I suggest the following title: Oberisk Approach, but the authors can find a better one. There are still several revisions required in this section.
    1. Remove Figure-1 since it is in French. If the authors still think that the steps in this figure are really important, they can keep the explanation part after a revision (lines 84-100). I don't think it is vital since, per the authors' statement, "This canvas did not scale for real studies. It was removed from the final version 93
      of ANSSI’s guide."
    2. Adding a flowchart or another figure to explain the steps of Oberisk will be helpful for the reader to follow the approach (i.e., the stages of how Oberisk works). 
  4. Figure 2 is low resolution. I cannot see what is written there. Once the authors add the flowchart or figure I requested in (3.b), Figure-2 will be irrelevant. 
  5. Other minor issues:
    1. A paragraph, typically, is not less than four sentences. Revise the paper based on this. I provide a sample list (not exhaustive) below.
      • Merge the first and the second paragraphs (lines 26-27)
      • Line 54 - not a new paragraph, merge with the previous one
      • Line 95- not a new paragraph, merge with the previous one
      • Merge paragraphs in lines 880-881
    2. Line 875: should be COVID not CoViD
    3. In the Discussion section, the authors can discuss if Oberisk can be used for risk analysis methods other than EBIOS, such as OCTVE, ForTE, RRA, NIST, ISO.

Reviewer 2 Report

The submitted article presents a novel, free and relatively easy to use framework for collaborative elicitation of cybersecurity requirements with the respective tool in the form of a set of datasheets. They are designed and tested for use in face-to-face and online brainstorming sessions. The proposed article is an excellent fit to Information’s special issue on “Scientific Approaches to Requirements Engineering: Research, Applications, Education and Future Directions”. 

The proposed framework is comprehensive, covering all main issues in deciding on the mitigation of cybersecurity risks, and provides for adequate inclusion in the deliberations of all main groups of stakeholders. The authors provide a walkthrough the steps of the framework methodology, along with relevant examples from the application to a naval case.  

The article is clearly written and well structured. However, the authors do not provide outline of the article, usually included in the end of the introductory section.

More information on other relevant frameworks and tools will help set Oberisk in proper context.

Higher quality of some of the figures is needed, in particular Figure 1, Figure 2, Figure A20. In fact, and in addition to being in French, Figure 1 does not add much to the description in the text and can be dropped out.

Figure 2 can hardly be provided in a readable version. I would recommend to drop it as well and add the framework (Oberisk) and the data sheets also here as supplementary material. It would also be useful to include in the text or in the description of the supplementary material a table listing workshops and respective main/standard and alternative datasheets, using the actual file naming convention.

Figure 18 also seems redundant. It shows just an empty template – the same like the one used in Figure 17.

On pages 7-9 the authors do not follow consistently their own designation of “primary”, business” “process-type” (services and functions) assets, introducing for example “primary services” on line 321. This is somewhat confusing while reading the article, and might be an obstacle if one decides to use the framework and the data sheets.  

Figure 11 is confusing and not typical for a “radar chart” (used for more than two, usually five or more, dimensions). It seems that a typical two-dimensional chart with axes from “low” to “high” would allow the same quality of visualization (even though the cases of highest interest won’t be positioned in the center of the diagram).

It is unclear where the statement “Indeed, some 100% technical profiles are now tagged DevOps” (line 33) comes from.

In the concluding section the authors do not provide the typical “way ahead” discussion. Instead, they focus on the practical utility of the framework. 

The language is very good. Yet, another round of careful reding will be needed to avoid cases like:

  • “but… on the one hand, traveling and gathering in a room to discuss the topic has become difficult, if not impossible, and on the other hand …” in the abstract
  • “These obviously include…” line 52 – the use if qualitive descriptions here and elsewhere
  • “CoViD” - unusual capitalization
  • “… allowed internal and external stakeholders [to] see” (line 293)
  • “Beyond, one should consider postponing the analyses” (lines 495-496)
  • “The diagram comprehends two axes” (lines 541-542)

More importantly, there are several terminological issues in the use of established risk management terms, i.e.:

  • The use of “feared [events]” (line 89 and below, e.g., section 3.1.8) in regard to risk assessment is unusual in the English language literature. More often authors discuss “likely/plausible high-impact events” or (according to ISO) “events that might create, enhance, prevent, degrade, accelerate or delay the achievement of objectives”
  • “risk likelihood assessment” (line 447) – the assessment of likelihood (of an event) is part of risk assessment.
  • “rejected risks” (line 599, line 623) – in fact, the authors mean events related to a low risk, which can be accepted (i.e., we understand that it exists, but decide to do nothing about it). The respective “strategic scenarios” can indeed be rejected (line 615), i.e., not to be taken into account in further deliberations.
  • “risk probability scale” (on figure 15, line 763) – Figure 15 shows the probability, or rather the likelihood of the scenario, not the risk

Round 2

Reviewer 1 Report

The authors did a good job of revising the manuscript. They addressed most of the comments in the first-round reviews. I suggest the following edits.

  1. Authors can cite more scholarly articles. I provide a list of survey articles related to cyber risk analysis, which can be used directly to contribute to the manuscript or through the papers they cited.
    • Böhme, R., Laube, S., & Riek, M. (2019). A fundamental approach to cyber risk analysis. Variance12(2), 161-185.
    • Fenz, S., Heurix, J., Neubauer, T., & Pechstein, F. (2014). Current challenges in information security risk management. Information Management & Computer Security.
    • Gritzalis, D., Iseppi, G., Mylonas, A., & Stavrou, V. (2018). Exiting the risk assessment maze: A meta-survey. ACM Computing Surveys (CSUR)51(1), 1-30.
    • McShane, M., Eling, M., & Nguyen, T. (2021). Cyber risk management: History and future research directions. Risk Management and Insurance Review24(1), 93-125.
    • Wangen, G., Hallstensen, C., & Snekkenes, E. (2018). A framework for estimating information security risk assessment method completeness. International Journal of Information Security17(6), 681-699.
    • Shameli-Sendi, A., Aghababaei-Barzegar, R., & Cheriet, M. (2016). Taxonomy of information security risk assessment (ISRA). Computers & security57, 14-30.
    • Nespoli, P., Papamartzivanos, D., Mármol, F. G., & Kambourakis, G. (2017). Optimal countermeasures selection against cyber attacks: A comprehensive survey on reaction frameworks. IEEE Communications Surveys & Tutorials20(2), 1361-1396.

Author Response

We thank the reviewer for these contributions.

To address this comment we revised and implemented the literature review section

Reviewer 2 Report

The authors have reflected most of my suggestions from the first review round and provided convincing explanation for the cases when they decided not to change the original text.

I recommend to publish this article.

Another round of copy-editing of the final text will be needed prior to publication to avoid, for example, cases like "Neither will not be visible in the final report" (line 1061).

Author Response

We thank the reviewer for pointing out this issue.

We applied the necessary modifications and adjustments all along the paper.
